 **eLIFE**

# Sgol2 provides a regulatory platform that coordinates essential cell cycle processes during meiosis I in oocytes

Ahmed Rattani[1], Magda Wolna[1†], Mickael Ploquin[1†‡a], Wolfgang Helmhart[1†‡b], Seamus Morrone[2], Bernd Mayer[3], Jonathan Godwin[1], Wenqing Xu[2], Olaf Stemmann[3], Alberto Pendas[4], Kim Nasmyth[1*]

[1]Department of Biochemistry, University of Oxford, Oxford, United Kingdom; [2]Department of Biological Structure, University of Washington, Seattle, United States; [3]Department of Genetics, University of Bayreuth, Bayreuth, Germany; [4]Campus Miguel de Unamúno, Instituto de Biología Molecular y Celular del Cáncer (CSIC-USAL), Salamanca, Spain

**\*For correspondence:** kim.nasmyth@bioch.ox.ac.uk

†These authors contributed equally to this work.

‡**Present address:** [a]New England Biolabs, Évry, France; [b]Institute of Molecular Life Sciences, University of Zürich, Zürich, Switzerland

**Competing interests:** The authors declare that no competing interests exist.

**Reviewing editor**: Tony Hyman, Max Planck Institute of Molecular Cell Biology and Genetics, Germany

**Abstract** Accurate chromosome segregation depends on coordination between cohesion resolution and kinetochore-microtubule interactions (K-fibers), a process regulated by the spindle assembly checkpoint (SAC). How these diverse processes are coordinated remains unclear. We show that in mammalian oocytes Shugoshin-like protein 2 (Sgol2) in addition to protecting cohesin, plays an important role in turning off the SAC, in promoting the congression and bi-orientation of bivalents on meiosis I spindles, in facilitating formation of K-fibers and in limiting bivalent stretching. Sgol2's ability to protect cohesin depends on its interaction with PP2A, as is its ability to silence the SAC, with the latter being mediated by direct binding to Mad2. In contrast, its effect on bivalent stretching and K-fiber formation is independent of PP2A and mediated by recruitment of MCAK and inhibition of Aurora C kinase activity respectively. By virtue of its multiple interactions, Sgol2 links many of the processes essential for faithful chromosome segregation.

## Introduction

The production of haploid gametes from diploid germ cells depends on two rounds of chromosome segregation (meiosis I and II) without an intervening round of DNA replication. Defects during the first or second meiotic division in oocytes lead to formation of aneuploid eggs, which in humans occurs with a frequency between 10 and 30% and is a major cause of fetal miscarriages (*Hassold and Hunt, 2001*). Understanding the causes of meiotic chromosome missegregation will require clarifying not only the forces and regulatory mechanisms governing meiotic chromosome segregation but also how these are coordinated.

At the heart of this process are two opposing forces. The pulling forces produced by kinetochore-microtubules attachments (K-fibers) and resisting forces generated by sister chromatid cohesion, which counteracts K-fiber forces if and when kinetochores attach to microtubules with different polarities. During meiosis I, cohesion along chromosome arms holds bivalent chromosomes together following the creation of chiasmata produced by reciprocal recombination between homologous non-sister chromatids. This cohesion must persist during the attachment of maternal and paternal kinetochores to microtubules from opposite poles (bi-orientation) and resist the resulting spindle forces. The degree of traction exerted by meiotic spindles must create sufficient tension to facilitate bi-orientation (*Tachibana-Konwalski et al., 2010*). During the bi-orientation process, the initial attachment of kinetochores to the surface of the microtubule lattice (lateral attachment) is converted to attachments

**eLife digest** Human reproductive cells—eggs and sperm—are produced through a process called meiosis. This means a 'parent' cell in the ovaries or testes undergoes two stages of cell division: it first divides into two cells, which then divide again to produce four 'daughter' cells. A crucial part of meiosis is ensuring that each daughter cell has half the number of chromosomes that the parent cell did.

Before the first round of meiosis, the chromosomes in the parent cell are copied to produce enough chromosomes for the four daughter cells. The distribution of these chromosomes between the daughter cells is determined by the opposing forces acting on them. The pairs of identical chromosomes produced during the copying process are held together by a proteinaceous glue, while microtubules attached to a structure called the spindle—which has poles at opposite ends of the cell—try to pull these pairs of chromosomes apart.

Rattani et al. now show that a protein called shugoshin-like 2 (Sgol2), which is involved in holding pairs of identical chromosomes together after the first round of division, has a bigger role than was previously realised. Sgol2 performs three other functions: it helps to align the chromosomes prior to division by, it is thought, facilitating the formation of the K-fibers that attach the microtubules to the chromosomes; it turns off the checkpoint that monitors the alignment of the chromosomes and the attachment of the microtubules; and it regulates a number of the enzymes involved in the process. The specific interactions that allow Sgol2 to perform these diverse functions in meiosis were also identified. Thus, Rattani et al. show that in linking so many essential processes throughout meiosis, Sgol2 appears to have a key, if not unique, role in determining the fate of chromosomes as cells divide.

to their plus ends (end on attachment), creating K-fibers. Because this process is intrinsically error prone, inappropriate attachments, for example those that connect maternal and paternal kinetochores to the same pole, must be disrupted through phosphorylation of kinetochore proteins by Aurora B/C kinases (*Kitajima et al., 2011*). However, because these kinases disrupt K-fibers, they must subsequently be down regulated once bivalents bi-orient correctly.

The first meiotic division is eventually triggered by activation of a gigantic ubiquitin protein ligase called the anaphase-promoting complex or cyclosome (APC/C) whose destruction of securin and cyclin B activates a thiol protease called separase that cleaves the kleisin subunit of the cohesin complex holding sister chromatids together (*McGuinness et al., 2005*; *Kudo et al., 2006*). This process converts chromosomes from bivalents to dyads. It is delayed until all bivalents have bi-oriented by the production, at kinetochores that have not yet come under tension, of a potent inhibitor of the APC/C called the mitotic checkpoint complex (MCC) whose Mad2 subunit binds tightly to the APC/C's Cdc20 co-activator protein. This regulatory mechanism, called the spindle assembly checkpoint (SAC), must be turned off before APC/C$^{Cdc20}$ can direct destruction of securin and cyclin B and thereby activate separase. Another pre-condition for cleavage, at least in yeast, is phosphorylation of cohesin's kleisin subunit by a pair of protein kinases, namely CK1δ/ε and DDK (*Ishiguro et al., 2010*; *Katis et al., 2010*). During the first meiotic division, cohesin is phosphorylated along chromosome arms but not at centromeres, which ensures that only cohesion along arms is destroyed by separase at the onset of anaphase I. The consequent persistence of cohesion at centromeres promotes bi-orientation of dyads during meiosis II.

Centromeric cohesin avoids phosphorylation and therefore cleavage during the first meiotic division because separase activation is preceded by the recruitment to centromeres of orthologues of the *Drosophila* MEI-S332 protein, called shugoshins. Members of this family contain a conserved C-terminal basic region and an N-terminal homodimeric parallel coiled coil, which provides a docking site for protein phosphatase 2A's (PP2A) C and B' subunits (*Xu et al., 2009*). In budding yeast, mutant proteins defective specifically in PP2A binding fail to confer protection of centromeric cohesion during meiosis I (*Xu et al., 2009*).

Mammals have two members of the shugoshin family: Shugoshin-like protein 1 (Sgol1), and Shugoshin-like protein 2 (Sgol2). The former prevents centromeric cohesin from a process called the 'prophase pathway' that removes cohesin from chromosomes by a non-proteolytic mechanism soon

after cells enter mitosis (*McGuinness et al., 2005*; *Liu et al., 2013*). Sgol2, on the other hand, protects centromeric cohesin from separase at the first meiotic division (*Lee et al., 2008*; *Llano et al., 2008*). Sgol2's coiled coil domain binds PP2A in vitro (*Xu et al., 2009*) but whether this is vital for protecting centromeric cohesion is not known. Sgol2 also interacts with MCAK (*Huang et al., 2007*), a microtubule depolymerizing kinesin, implicated in correcting inappropriate kinetochore-microtubule interactions (error correction), and with Mad2 an essential component of the MCC (*Orth et al., 2011*). Shugoshins, though not explicitly Sgol2, have also been implicated in recruiting to kinetochores the Aurora B kinase, necessary both for the SAC and for error correction (*Tsukahara et al., 2010*; *Yamagishi et al., 2010*). Based on its interactions, Sgol2 has been linked to cohesion protection, the spindle assembly checkpoint, and error correction pathways. However, the physiological significance of these multiple interactions remain unclear.

We show here that Sgol2 defective in PP2A binding fails to protect centromeric cohesin, as found for Sgo1 in yeast (*Xu et al., 2009*). However, if this were the sole function of Sgol2, then chromosome behavior during meiosis I should be unaffected. To our surprise, we found that Sgol2 deficiency caused striking changes in chromosome and microtubule dynamics. It delayed bi-orientation of bivalents on meiosis I spindles, caused increased bivalent stretching, and greatly increased Aurora B/C kinase activity at kinetochores, which was accompanied by an increase in lateral and a decrease in end on kinetochore-microtubules attachments. Lastly, Sgol2 deficiency delayed considerably APC/C^Cdc20 activation, suggesting that it was required to shut off the SAC. Sgol2 helps to turn off the SAC by binding directly both PP2A and the MCC protein Mad2, it moderates chromosome stretching by recruiting to kinetochores the kinesin MCAK, and likely promotes formation of K-fibers by down-regulating the activity of Aurora B/C kinase specifically at kinetochores. Meiotic chromosome segregation is a highly complex process dependent on an array of different biochemical processes. Our findings imply that through its multi-domain structure, Sgol2 has an important if not unique role in coordinating many of the key processes within this array.

## Results

### Sgol2 interacts with PP2A through conserved residues in its coiled coil

An essential feature of shugoshins is a conserved N-terminal homodimeric coiled coil that binds PP2A (*Figure 1A*). Unlike Sgol1, which only interacts stably with the PP2A holoenzyme (a complex of A, B' and C subunits), Sgol2 also associates with a core complex containing only A and C subunits (*Xu et al., 2009*). To create versions of Sgol2 defective specifically in binding PP2A, conserved residues (namely L51, N54, R56, and E82) predicted to interact directly with PP2A were substituted by alanine or in the case of E82 also by lysine (*Figure 1B*). Likewise, the highly conserved asparagine (N55 on Sgol2), corresponding to N61 in Sgol1, which forms a hydrogen bond within the coiled coil enabling it to adopt a conformation compatible with PP2A binding, was substituted by isoleucine (*Figure 1B*). Importantly, binding between purified Flag- and His-tagged Sgol2 fragments spanning the coiled coil (cc) region (31–117) was unaffected by any of these mutations, implying that all permit coiled coil formation. A triple mutation L51A/N54A/R56A (3A) reduced dimerization modestly (*Figure 1D*) but this effect was not seen when the two proteins were co-expressed in bacteria (*Figure 1E*). To assess the effects on PP2A binding, GST-tagged PP2A A subunit was incubated with B' and C subunits in the presence of MBP-Sgol2cc-Flag. Pull down experiments revealed that N55I and the 3A triple mutation abolished co-purification of Sgol2 with the GST-tagged A subunit while N54A and R56A reduced it. In contrast, L51A and E82A had little or no effect while E82K caused a modest reduction (*Figure 1F*).

### Sgol2 protects centromeric cohesion by recruiting PP2A to kinetochores

To investigate whether association of PP2A with Sgol2's coiled coil is necessary for protecting centromeric cohesion, we compared the effect of the mutations on PP2A's co-localization with Sgol2 at kinetochores with their effect on the maintenance of cohesion after oocytes have undergone the first meiotic division. Sgol2 and PP2A co-localize at inner kinetochores of bivalent chromosomes at metaphase I in wild type oocytes (*Figure 2A*) but neither protein was detected at this location in *sgol2^Δ/Δ* oocytes (n = 26). Importantly, microinjection of wild type Sgol2 encoding mRNA into *sgol2^Δ/Δ* oocytes restored accumulation at kinetochores of both Sgol2 and PP2A at metaphase I. In contrast, microinjection of mRNAs carrying N54A, N55I, or 3A mutations restored Sgol2 but not PP2A (*Figure 2A*), implying

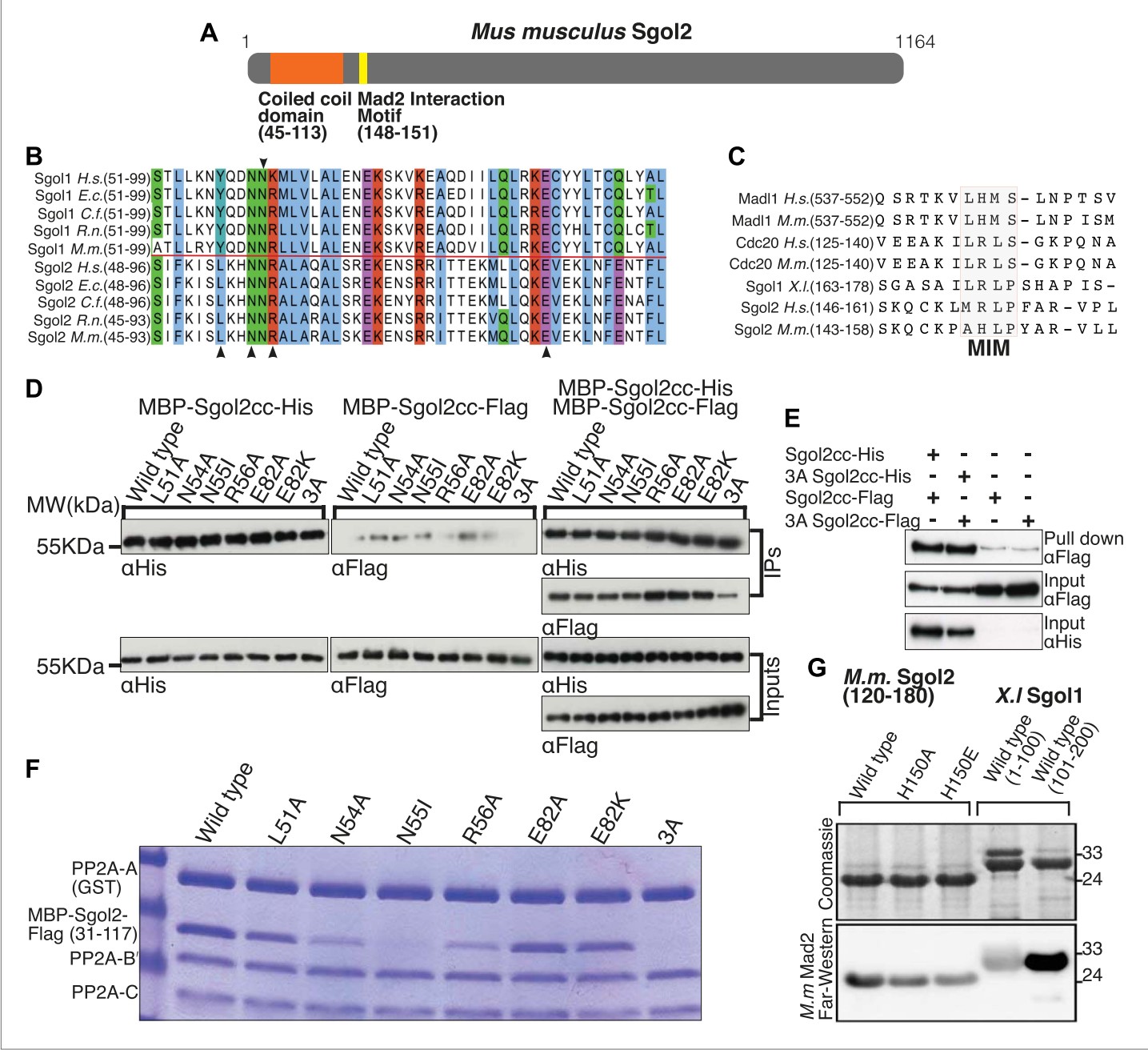

**Figure 1**. Sgol2 binds to PP2A and Mad2 through coiled coil domain and Mad2 Interaction Motif (MIM), respectively. (**A**) Schematic diagram indicating relative positions of the coiled coil domain and the MIM sequence in *M. musculus* Sgol2 (not drawn to scale). (**B**) Sequence alignment of the coiled coil domains of Sgol1 and Sgol2 from different vertebrate species. Arrowheads below the alignment indicate conserved residues (L51, N54, R56, and E82) in the coiled coil domain of Sgol2 mapping onto binding sites between PP2A and Sgol1. The conserved asparagine residue (N55 in mSgol2), important for maintaining the interaction surface of the coiled coil domain, is marked by an arrowhead above the alignment. (**C**) Sequence alignment of conserved MIM (Mad2 Interaction Motif) in Sgol2, Mad1, and Cdc20 from *Xenopus*, human and mouse origin. (**D**) Dimerization assay of coiled coil domains from wild type and mutant Sgol2. The Flag-tagged MBP-Sgol2cc (31–117) was pulled down by the His-tagged MBP-Sgol2cc (31–117) with the same substitution. Homodimerisation was assessed by the relative abundance of each form in the IP by western blots. (**E**) Co-immunoprecipitation assay of MBP-Sgol2cc from wild type and 3A mutant are shown to validate the homodimerisation of coiled coil domains. MBP-Sgol2cc-Flag and MBP-Sgol2cc-His were expressed either independently, or co-expressed. After His-tag pull-down, each fraction was separated on a 12% SDS-PAGE and analyzed by immunoblotting with anti-Flag and anti-His antibodies. Input was resolved on a 12% SDS-PAGE. (**F**) The in vitro binding assay of PP2A A(GST-tagged), B', and C subunits with wild type MBP-mSgol2cc-Flag and variants thereof. After GST pull down, eluates were resolved by SDS-PAGE and Coomassie stained. (**G**) Sgol2 and Mad2 binding assay. Sgol2 fragment spanning amino-acid (aa) residues 120–180 with wild type and mutant MIM sequences were assessed for Mad2 interaction by far-western blot. Fragments from *Xenopus* Sgol1, with and without MIM domain, were used as positive and negative controls.

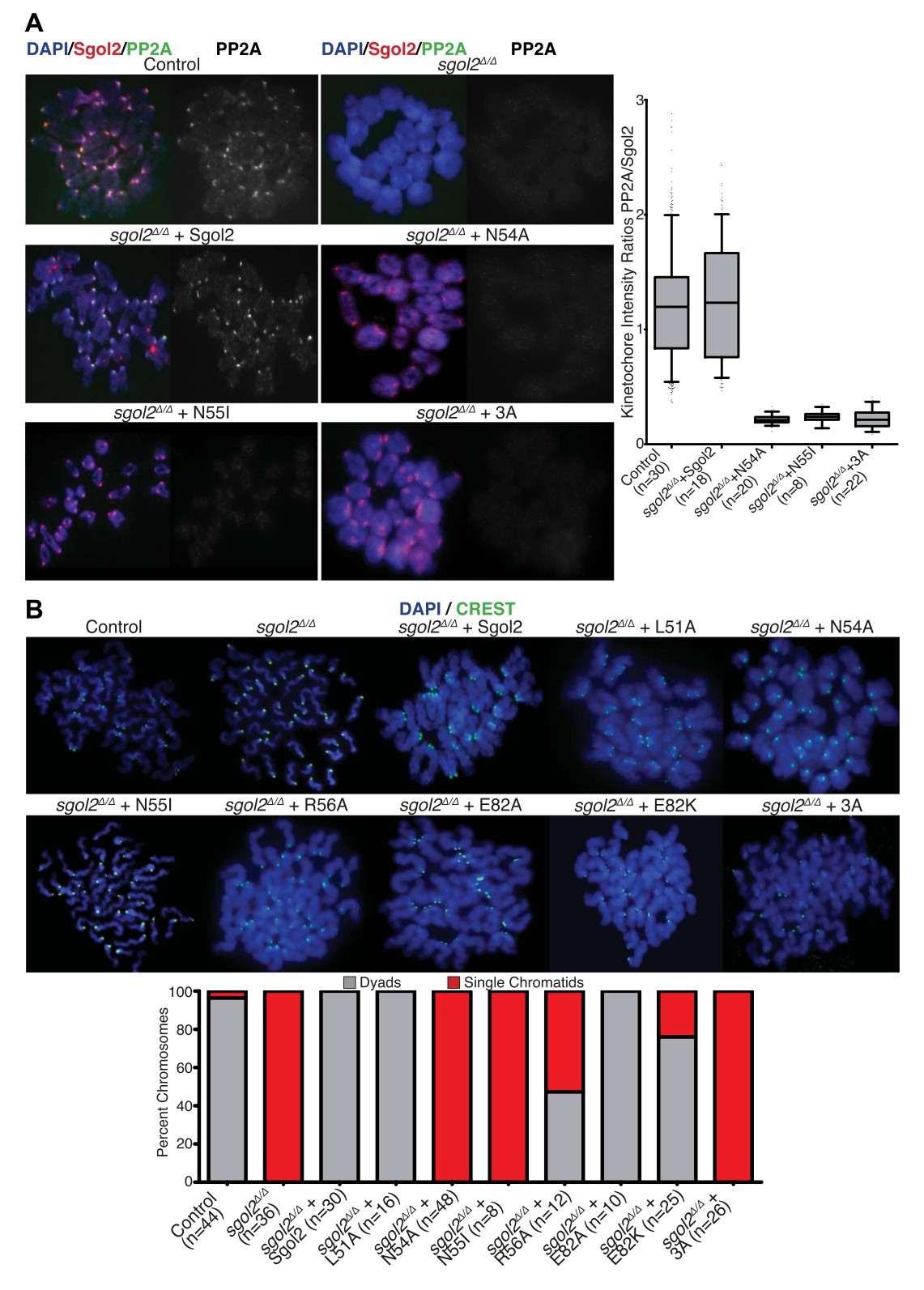

**Figure 2**. Sgol2-PP2A interaction is required to recruit PP2A to kinetochores and protect centromeric cohesion at the first meiotic division. (**A**) GV oocytes were harvested in the presence of IBMX. Microinjections were performed at GV stage in M2 medium supplemented with IBMX. Oocytes were then cultured in IBMX-free medium for about 6 hr, corresponding to metaphase I stage. Chromosome spreads were prepared and stained with DAPI

*Figure 2. Continued on next page*

Figure 2. Continued

(blue), Sgol2 (red), and PP2A-C (green). Box and whisker plot shows fluorescence intensity ratios of PP2A and Sgol2 at kinetochores. Upper and lower bars indicate 95th and 5th percentiles, respectively. Numbers of oocytes examined are indicated (n). (**B**) Oocytes harvested at GV stage were matured in M16 medium for up to 24 hr. Chromosome spreads prepared from oocytes that have extruded the first polar body were stained with DAPI (blue) to visualize DNA and CREST (green) to mark centromeres. Frequencies of dyads and single chromatids were quantified.

that binding of PP2A to Sgol2's coiled coil is necessary for recruiting the phosphatase to kinetochores in vivo. Chromosome spreads revealed that the failure of *sgol2*$^{\Delta/\Delta}$ oocytes to maintain cohesion between sister centromeres after meiosis I was reversed by microinjection of wild type but not N54A, N55I, or 3A mRNAs (*Figure 2B*). L51A and E82A, which had little or no effect on PP2A's binding to Sgol2's coiled coil in vitro, also fully restored cohesion protection while R56A and E82K, which caused a modest reduction in the Sgol2cc-PP2A interaction vitro, only partially restored it, producing metaphase II oocytes containing individual chromatids as well as dyads (*Figure 2B*). Lastly, we noticed that injection of 3A Sgol2 mRNAs also produced centromeric cohesion defects in wild type oocytes (*Figure 3A,B*). The correlation between the effects of mutations on binding PP2A in vitro and on protecting cohesion in vivo implies that the interaction of PP2A with Sgol2's coiled coil is essential for protecting cohesin from separase at the first meiotic division.

## Sgol2 silences the spindle assembly checkpoint by binding PP2A and Mad2

DIC video microscopy revealed that *sgol2*$^{\Delta/\Delta}$ oocytes extruded polar bodies on average 2 hr later than oocytes from wild type littermates. Moreover, only 61% of *sgol2*$^{\Delta/\Delta}$ oocytes extruded the first polar body compared with 88% in wild type controls (*Figure 4A*). To investigate this, wild type and *sgol2*$^{\Delta/\Delta}$ oocytes were microinjected with mRNA encoding histone H2B-mCherry to mark chromosomes and Securin-eGFP to measure APC/C$^{Cdc20}$ activity (*Figure 4B*). 40% of mutant oocytes failed to activate APC/C$^{Cdc20}$ compared to 12% in wild type controls (*Figure 4C*), and those that did so were delayed by about 2 hr (*Figure 4D*, *Figure 4—figure supplement 1*). Surprisingly, microinjection of Sgol2 mRNA not only increased the fraction of oocytes that activated APC/C$^{Cdc20}$ to wild type levels (89%), but also advanced activation so that it took place about 2 hr earlier than in wild type (*Figure 4E–G*, *Figure 4—figure supplements 2 and 3*). Because microinjection of mRNA encoding a version of Cdc20 (R132A) that cannot bind Mad2, but not wild type Cdc20, induced securin destruction in all *sgol2*$^{\Delta/\Delta}$ oocytes (*Figure 4C*), we conclude that Sgol2 facilitates APC/C$^{Cdc20}$ activation by silencing the SAC and that this process is dose dependent.

The recent observation that shugoshins, like Sgol2 from humans and Sgo1 from *Xenopus*, bind Mad2 in a manner similar to Mad1 and Cdc20 through a conserved Mad2 Interaction Motif (MIM) (*Orth et al., 2011*) provided a clue as to how Sgol2 might regulates the SAC. This motif is conserved in mouse Sgol2 (*Figure 1C*) and far-western blots confirmed that interaction between mSgol2 and Mad2 is weakened by substituting the basic amino acid at its center with alanine (H150A) or glutamic acid (H150E) (*Figure 1G*). Strikingly, H150E and H150A Sgol2 mRNAs were less effective than wild type in increasing the fraction of *sgol2*$^{\Delta/\Delta}$ oocytes that activated APC/C$^{Cdc20}$ (*Figure 4E*). They also failed to advance the timing of activation (*Figure 4F*, *Figure 4—figure supplement 2B,C*). Importantly, neither mutation affected Sgol2's association with kinetochores, its ability to recruit PP2A (*Figure 3C*) or MCAK (*Figure 5A*) to this location, or protection of centromeric cohesion (*Figure 3D*). Interestingly, N54A, which reduces PP2A binding, also abrogated Sgol2's ability to promote APC/C$^{Cdc20}$ activation (*Figure 4E,F*, *Figure 4—figure supplement 2A,D*) without having any effect on MCAK recruitment (*Figure 5A*). We conclude that Sgol2 has an important role in silencing the SAC during meiosis I by binding directly both to Mad2 and PP2A.

## Sgol2's recruitment of MCAK regulates bivalent stretching

We noticed that chiasmata appeared terminalized in a large fraction of bivalent chromosomes from *sgol2*$^{\Delta/\Delta}$ oocytes (*Figure 5A*). Because this effect was not accompanied by any reduction of cohesin along inter-chromatid axes (*Figure 5B*), it is unlikely to be caused by precocious Rec8 cleavage. Live cell imaging of oocytes microinjected at the GV stage with mRNAs encoding H2B-mCherry and GFP-MCAK showed that maternal and paternal kinetochores were pulled further apart in *sgol2*$^{\Delta/\Delta}$ oocytes at metaphase I (*t* test, p<0.0001). In other words, bivalents were stretched more and possibly the

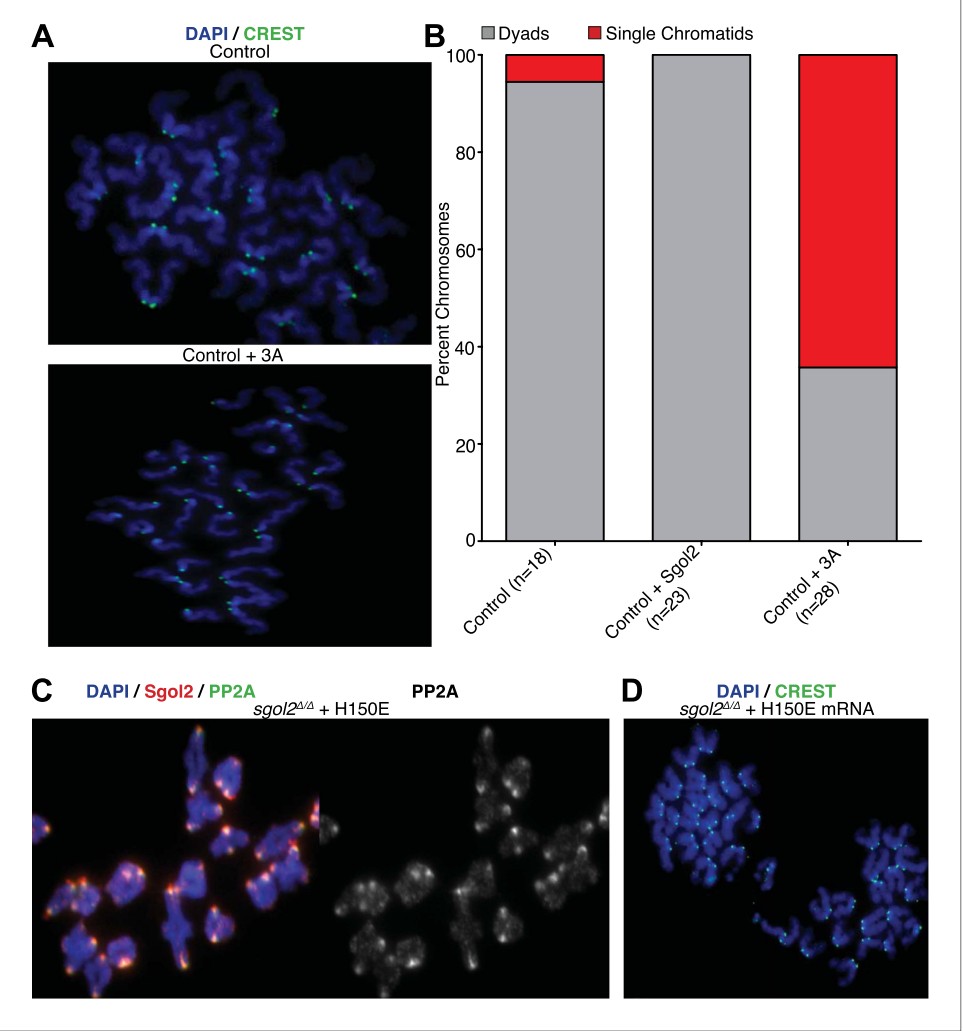

**Figure 3**. Sgol2-PP2A, but not Sgol2-Mad2, interaction is required to protect centromeric cohesion at the first meiotic division. (**A**) Oocytes harvested at GV stage from wild type (control) mice were microinjected with wild type or 3A mutant Sgol2 mRNA in M2 medium supplemented with IBMX. After releasing the oocytes in M16 medium without IBMX, chromosome spreads were prepared from metaphase II-arrested oocytes and stained with DAPI (blue) and CREST (green). (**B**) Frequencies of dyads and single chromatids observed on metaphase II-stage chromosome spreads. Numbers of oocytes examined are indicated (n). (**C**) Metaphase I localization of Sgol2 (red) and PP2A-C (green) on chromosomes from sgol2$^{\Delta/\Delta}$ oocytes microinjected with H150E Sgol2 mRNA. (**D**) DAPI (blue) and CREST (green) stained chromosome spreads prepared from metaphase II-stage oocytes.

stretching of distal parts of bivalent chromosomes resulted in increased separation of homologous pairs on chromosome spreads. Importantly, increased kinetochore separation in sgol2$^{\Delta/\Delta}$ oocytes was invariably observed on bivalents that had bi-oriented (*Figures 5C and 6*; see also *Videos 1 and 2*), but was not accompanied by any change in the overall length of metaphase I spindles (ANOVA, p=0.97; *Figure 5D*). Interestingly, microinjection of Sgol2 mRNA not only rescued chiasmata structure (*Figure 5A*), but also reduced inter-kinetochore distances to below that seen in wild type (*Figures 5C and 6*; *Video 3*, *t* test p<0.0001), without altering spindle length (*Figure 5D*). Because a similar reduction in inter-kinetochore distances was observed with N54A or H150E mutations (*Figures 5C and 6*; *Videos 4 and 5*, *t* test p<0.0001), regulation of bivalent stretching by Sgol2 does not depend on its ability to bind PP2A or Mad2. Thus, the effect cannot be due to changes either in arm cohesion protection or duration of metaphase I.

Inter-kinetochore distances correlated inversely with GFP-MCAK signals at kinetochores. Thus, MCAK is missing from kinetochores in sgol2$^{\Delta/\Delta}$ oocytes and more abundant (even than wild type) in mutant cells microinjected with Sgol2 mRNA (*Figures 5A,C and 6*). Because phosphorylation of human Sgol2 at

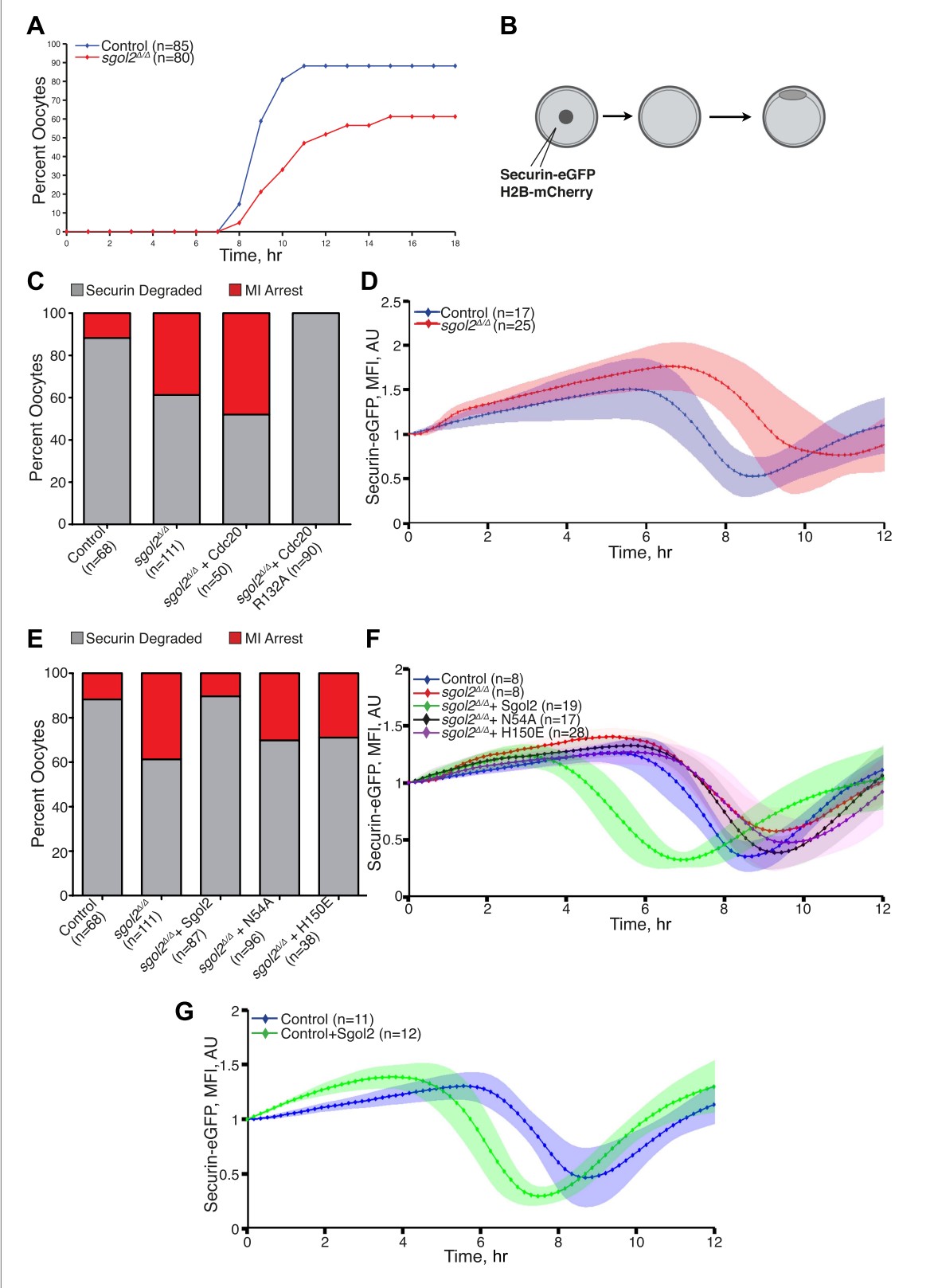

**Figure 4**. Sgol2 silences spindle assembly checkpoint through its interactions with PP2A and Mad2. (**A**) Kinetics of the first polar body extrusion (PBE) of oocytes cultured in vitro. Numbers of oocytes examined are indicated (n). (**B**) Oocytes were microinjected with indicated mRNAs at the GV stage. Chromosome movements and kinetics of Securin-eGFP were visualized by time-lapse confocal microscopy. (**C**) Frequencies of oocytes that destroyed

*Figure 4. Continued on next page*

*Figure 4. Continued*

Securin-eGFP, hence activated APC/C$^{Cdc20}$, at the metaphase-anaphase transition are shown. SAC was inactivated by expressing a dominant-negative version of the APC/C activator Cdc20 (Cdc20–R132A) that cannot be bound by Mad2. (**D**) Data represent the mean and standard deviations of Securin-eGFP mean fluorescence intensity levels in *sgol2*$^{Δ/Δ}$ (red) and litter mate control (blue) oocytes at each time point. Values from individual oocytes were normalized relative to that at GVBD (0 hr), and mean and standard deviations of the population are plotted in arbitrary units, (AU), against time. Securin-eGFP curves from individual oocytes are shown in *Figure 4—figure supplement 1*. (**E**) Securin-eGFP destruction was used as a marker for APC/C activation. Control and *Sgol2* knockout are compared to *sgol2*$^{Δ/Δ}$ oocytes microinjected with indicated Sgol2 mRNA. (**F**) GV-normalized mean and standard deviations of Securin-eGFP levels are plotted against time for the indicated groups of oocytes. Securin-eGFP curves from individual oocytes are displayed in *Figure 4—figure supplement 2*. (**G**) Microinjection of Sgol2 mRNA accelerates APC/C$^{Cdc20}$ activation in wild type oocytes. Mean and standard deviations of the time course measurements of Securin-eGFP mean fluorescence intensity are displayed. Securin-eGFP curves from individual oocytes are shown in *Figure 4—figure supplement 3*.

The following figure supplements are available for figure 4:

**Figure supplement 1**. Time course measurements of Securin-eGFP mean fluorescence intensity in control and *Sgol2* knockout oocytes.

**Figure supplement 2**. Comparison of Securin-eGFP mean fluorescence intensity curves in Sgol2 knockout oocytes and *sgol2*$^{Δ/Δ}$ oocytes expressing the indicated form of *Sgol2*$^{Δ/Δ}$ mRNA.

**Figure supplement 3**. Microinjection of Sgol2 mRNA accelerates APC/C$^{Cdc20}$ activation in wild type oocytes.

T537 and T620 (corresponding to T521 and T600 on *M.m* Sgol2) by Aurora B has been implicated in the association of MCAK with Sgol2 in mitotic cells (*Tanno et al., 2010*), we tested the effect of AZD1152, an Aurora B (and C) kinase inhibitor. This eliminated accumulation at kinetochores of both T521 phosphorylated Sgol2 (*Figure 7B*) and MCAK (*Figure 5A*, *Figure 5—figure supplement 1*). Crucially, injection of mRNA encoding T521A T600A Sgol2 into *sgol2*$^{Δ/Δ}$ oocytes failed to restore accumulation of MCAK at kinetochores (*Figure 5A*, *Figure 5—figure supplement 1*) and to reduce inter-kinetochore distances (*Figures 5C and 6*; *Video 6*, *t* test p=0.027). Interestingly, neither the Aurora B/C inhibitor nor T521A T600A prevented MCAK's recruitment to chromosome arms (*Figure 5—figure supplement 1*). The effect of T521A T600A on MCAK's kinetochore binding was highly specific as the mutant Sgol2 protein still accumulated at kinetochores (albeit to a lesser extent), recruited PP2A to this location, and protected centromeric cohesion from separase (*Figure 5E,F*).

These observations suggest that bivalent stretching might be inhibited by recruitment of MCAK to kinetochores by Sgol2. If so, increased inter-kinetochore distances in *sgol2*$^{Δ/Δ}$ oocytes should be reduced by targeting MCAK to centromeres by artificial means, namely by expression of an MCAK protein whose N-terminus is fused to the C-terminus of CENPB-eGFP. Remarkably, CENPB-eGFP-MCAK, but not CENPB-eGFP, reduced inter-kinetochore distances (*Figure 5C*, *Figure 5—figure supplement 2*; *Video 7*, *t* test p<0.0001). We conclude that Sgol2 moderates the pulling force exerted on bivalents by recruiting MCAK to kinetochores, a process dependent on phosphorylation of Sgol2 by Aurora B/C kinases.

## Sgol2 promotes chromosome congression and bi-orientation

The period between Germinal Vesicle Breakdown (GVBD) and anaphase can be divided according to the state of spindles and chromosomes into four distinct phases (*Schuh and Ellenberg, 2007*; *Kitajima et al., 2011*). During phase 1, multiple microtubule-organizing centers (MTOCs) form a ball of polymerizing microtubules on whose surface chromosomes are distributed individually. This is followed by a period (phase 2) in which chromosomes slide laterally over the ball shaped spindle apparatus, eventually congressing to form a 'prometaphase' belt around it. During phase 3, chromosomes invade the cluster of microtubules, which elongates along an axis perpendicular to the earlier chromosome belt, forming a barrel-shaped bipolar spindle. This highly dynamic phase culminates when chromosomes bi-orient on a metaphase plate within the bipolar spindle. During the fourth and final stage (phase 4), bi-oriented chromosomes oscillate around the metaphase plate with low amplitudes (Phase 4).

Live-cell confocal imaging of wild type mouse oocytes microinjected with H2B-mCherry and GFP-MCAK showed that MCAK accumulates at MTOCs and on microtubules during phase 1 (*Figure 6*; *Video 1*) but does not appear at kinetochores until chromosomes congress on the surface of the ball at the start of phase 2. This event is accompanied by an increase in Sgol2's abundance at kinetochores

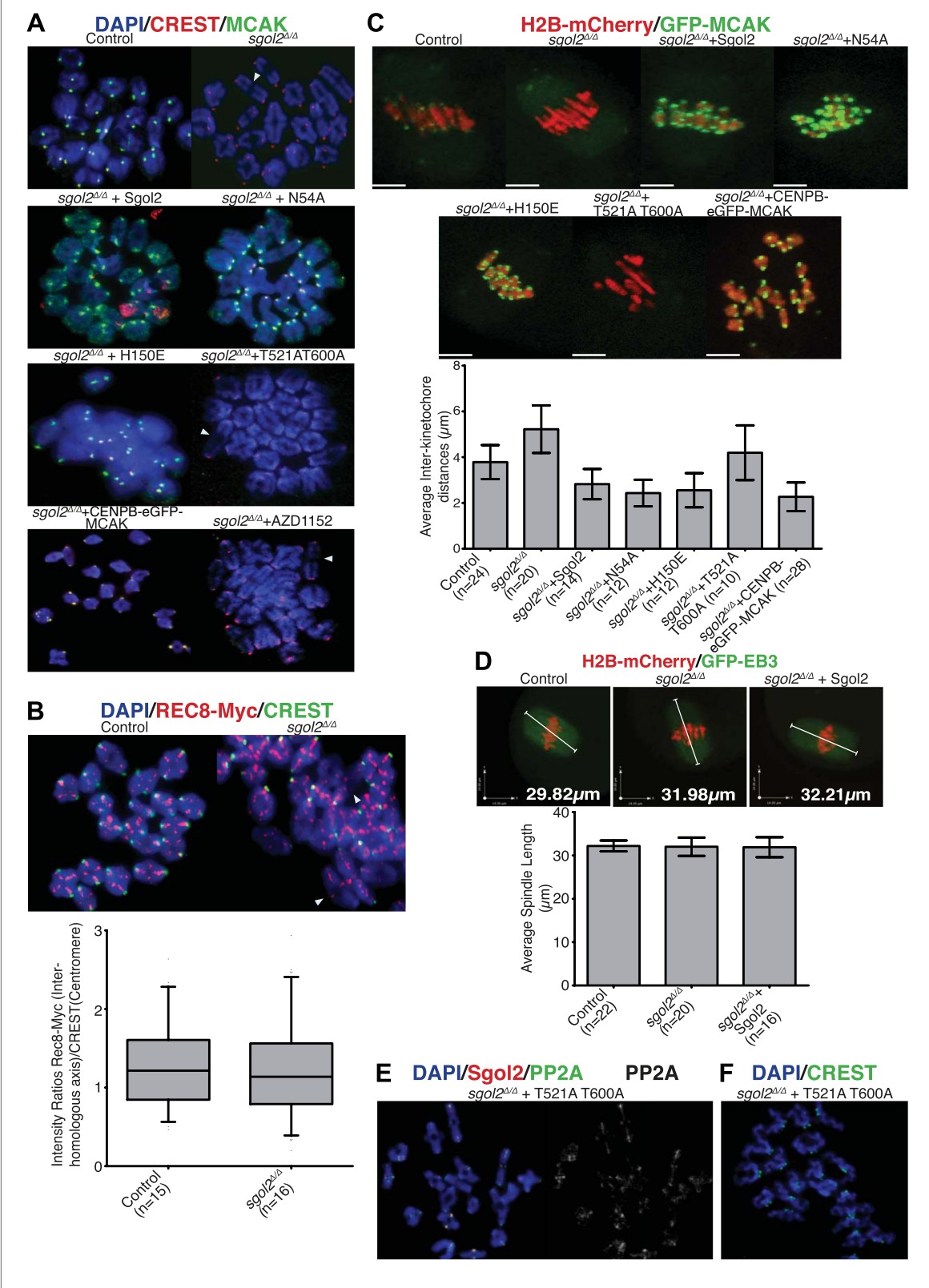

**Figure 5**. Sgol2, by recruiting MCAK to kinetochores, maintains inter-kinetochore distances between homologous chromosomes. (**A**) GV stage oocytes were harvested and microinjected in M2 medium supplemented with IBMX. Release from IBMX resulted in GVBD and at 6–7 hr post-GVBD chromosome spreads were prepared. Slides were stained with DAPI (blue), CREST (Red), and MCAK (green). Arrowheads indicate distally attached bivalent

*Figure 5. Continued on next page*

*Figure 5. Continued*

chromosomes. (**B**) *Sgol2* targeted mice were crossed to *(Tg)Rec8-Myc* females to generate control and *sgol2*$^{\Delta/\Delta}$ expressing the *Rec8-myc* BAC transgene. Chromosome spreads were prepared at 6–7 hr post-GVBD and were stained with DAPI (blue), CREST (green), and c-Myc (Red). Intensity ratios of Rec8-Myc (on inter-kinetochore axes) and CREST (at centromeres) were calculated. (**C**) Representative Z-projected confocal microscopy images and average (±SDs) inter-kinetochore distance measurements from metaphase I stage oocytes expressing H2B-mCherry (chromosomes, red) and GFP-MCAK (green). Scale bars represent 5 μm. (**D**) Z-projected (12 slices 1.5 μm apart) live-cell confocal images and spindle length measurements (±SDs) from metaphase I stage oocytes expressing EB3-GFP (microtubule plus ends, green) and H2B-mCherry (chromosomes, red). Scale bar is 14 μm. (**E**) Metaphase I localization of Sgol2 (red) and PP2A-C (green) on chromosomes from *sgol2*$^{\Delta/\Delta}$ oocytes microinjected with T521A T600A Sgol2 mRNA. (**F**) DAPI (blue) and CREST (green) stained chromosome spreads prepared at metaphase II-stage from *sgol2*$^{\Delta/\Delta}$ oocytes microinjected at GV stage with T521A T600A Sgol2 mRNA.

The following figure supplements are available for figure 5:

**Figure supplement 1**. GV oocytes were harvested in M2 medium supplemented with IBMX.

**Figure supplement 2**. Oocytes harvested at GV stage were microinjected with either CENPB-eGFP or CENPB-eGFP fused to MCAK mRNA.

and by phosphorylation of T521 (*Figure 7A,B*). MCAK's abundance at kinetochores gradually increases during phase 3 as chromosomes invade the spindle apparatus and peaks as chromosomes bi-orient on the metaphase plate (*Figure 6*, top panel; *Video 1*). This sequence of events is substantially altered in *sgol2*$^{\Delta/\Delta}$ oocytes. Though GFP-MCAK co-localizes with MTOCs and microtubules, it never subsequently accumulates at kinetochores (*Figure 6*). Chromosomes associate with the surface of the ball, albeit more loosely than in wild type, and undergo visible stretching instead of forming a prometaphase belt (*Figures 6 and 8A*, *Figure 6—figure supplement 1*; *Video 2*). This excessive stretching becomes even more pronounced upon formation of bipolar spindles. Thus, the clear demarcation of congression during phase 2 and bi-orientation during phase 3 was largely absent. Microinjection of Sgol2 mRNAs into *sgol2*$^{\Delta/\Delta}$ oocytes greatly accelerated both invasion of the ball by chromosomes and their bi-orientation, with the latter occurring even earlier than in wild type oocytes and before metamorphosis of the ball into a bipolar spindle (*Figures 6 and 8A*, *Figure 6—figure supplement 1*; *Video 3*). Thus, while loss of Sgol2 delays bi-orientation, the modest over-expression caused by mRNA microinjection accelerates this process.

Crucially, the precocious bi-orientation caused by Sgol2 mRNA injection was unaffected by N54A or by H150E mutations but abrogated by T521A T600A (*Figures 6 and 8A*, *Figure 6—figure supplement 1*; *Videos 4, 5, 6*), which also abolished MCAK's accumulation at kinetochores and compromised chromosome alignment (*Figures 5A and 6*, *Figure 6—figure supplement 1*), possibly due to increased association of MCAK with the arms of bivalent chromosomes (*Figure 5—figure supplement 1*). Injection of Sgol2 T521A T600A still advanced polar body extrusion (*Figure 6*), demonstrating that SAC silencing by Sgol2 does not depend on rapid or possibly even accurate chromosome bi-orientation and suggesting that these activities are mediated by separate domains within Sgol2. Surprisingly, injection of mRNAs encoding CENPB-eGFP-MCAK instead of Sgol2 did not accelerate chromosome bi-orientation (data not shown), raising the possibility that Sgol2 facilitates bi-orientation by a mechanism either independent of or in addition to MCAK's recruitment to kinetochores.

## Sgol2 regulates kinetochore-microtubule interactions

Because, bi-orientation of bivalent chromosomes correlates with the appearance of kinetochore-microtubule bundles (*Kitajima et al., 2011*), we visualized these in oocytes fixed in 1% PFA following a brief treatment with a calcium-containing buffer, which destabilizes the more dynamic non-kinetochore-microtubules. This revealed three differences between wild type and *sgol2*$^{\Delta/\Delta}$ oocytes at 6 hr post-GVBD, when most kinetochores in wild type have relinquished lateral and formed end on attachments and chromosomes have bi-oriented. Kinetochore associated microtubules bundles (K-fibers) were thinner and more sparse, lateral attachments were greatly increased in frequency, and kinetochores (as detected by CREST antibodies) were frequently highly stretched in mutant oocytes (*Figure 8B*). Microinjection of Sgol2 mRNA had the opposite effect, eliminating lateral attachments and kinetochore stretching while greatly increasing the abundance of end-on microtubules associated with each kinetochore, an effect abrogated by T521A T600A (*Figure 8B*). This suggests that the accumulation at kinetochores of Sgol2 phosphorylated by Aurora B/C kinase stimulates the conversion of lateral to end

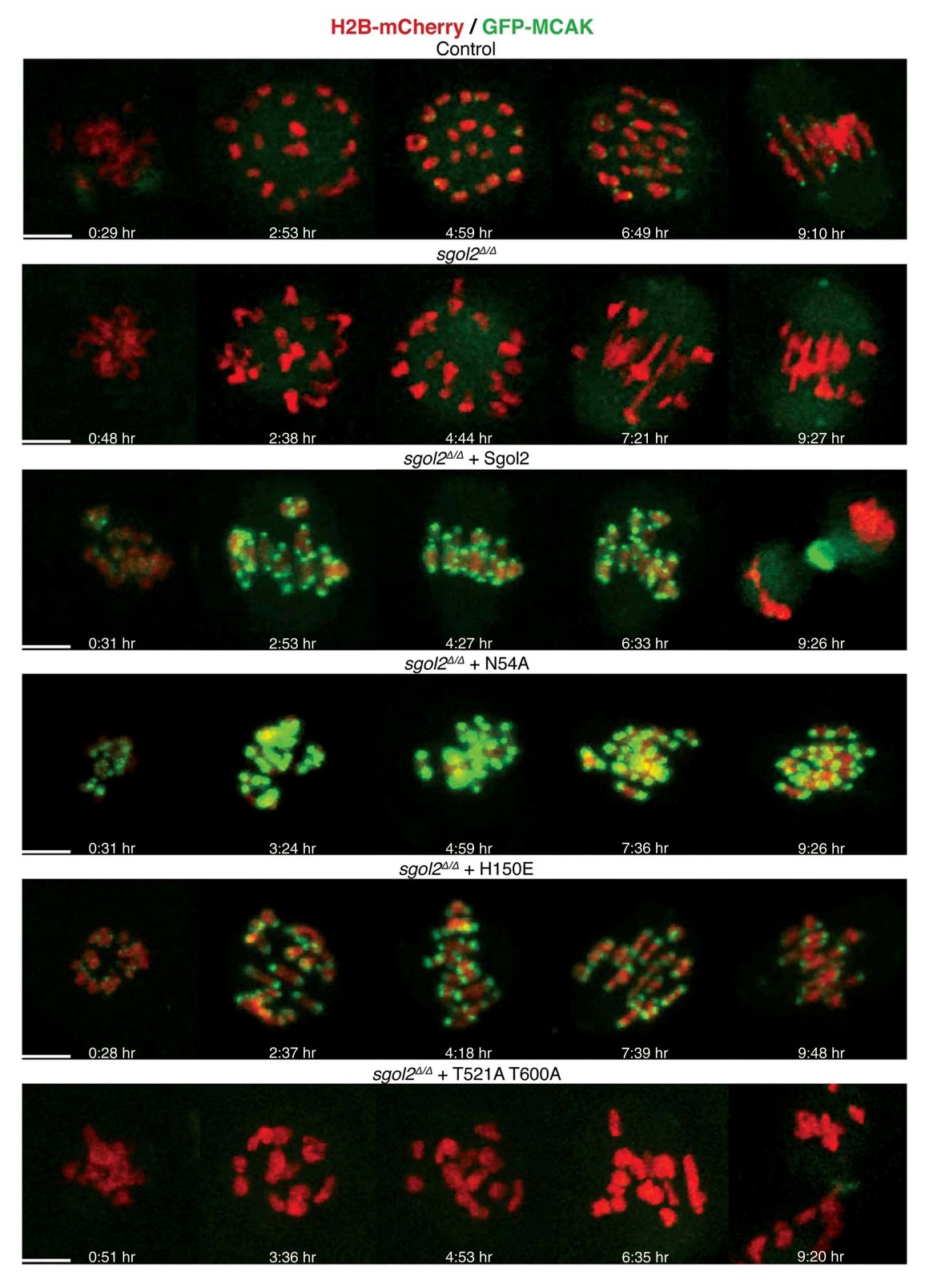

**Figure 6**. Sgol2 assists in bi-orientation of bivalent chromosomes on the metaphase plate. Time-lapse confocal microscopy images from oocytes expressing H2B-mCherry and GFP-MCAK. Representative images from litter-mate control, $sgol2^{\Delta/\Delta}$, $sgol2^{\Delta/\Delta}$ microinjected with wild type, N54A, H150E,

*Figure 6. Continued on next page*

*Figure 6. Continued*

and T521 T600A Sgol2 mRNA are aligned to show the dynamics of MCAK localization and chromosome movements along the meiotic spindle. GVBD normalized time (in hr) is indicated on each frame. Scale bars represent 5 μm.

The following figure supplements are available for figure 6:

**Figure supplement 1**. Live cell confocal microscopy images from oocytes expressing H2B-mCherry (chromosomes, red) and EB3-GFP (microtubules, green) are shown.

on attachments and that this may be the mechanism by which Sgol2 accelerates chromosome bi-orientation. Because injection of CENPB-eGFP-MCAK mRNAs instead of Sgol2 largely failed to increase end-on attachments at the expense of lateral ones (***Figure 8B***), we suspect that Sgol2 phosphorylated by Aurora B/C has a function at kinetochores besides MCAK recruitment (see below).

### Sgol2 inhibits Aurora B/C kinase activity at kinetochores

Because Aurora B kinase destabilizes kinetochore-microtubule attachments, at least partly by phosphorylating proteins within the KMN network (***Welburn et al., 2010***), we tested whether Sgol2 altered the phosphorylation of Ser24 on Knl1 (p-Knl1). Remarkably, the ratio of p-Knl1 to CREST signals in $sgol2^{\Delta/\Delta}$ oocytes at 6 hr post-GVBD was more than double that of wild type controls (***Figure 9A***). Microinjection of Sgol2 mRNA reduced this to below wild type, an effect unaltered by N54A but largely abolished by T521A T600A (***Figure 9A***). Similar results were obtained using a phospho-specific antibody against Ser100 on Dsn1 (data not shown) and one that detects activating auto-phosphorylation of Aurora A/B/C at Thr288, Thr232, or Thr198 (p-Aurora), respectively (***Figure 9B***). Interestingly, these effects were specific to kinetochores as auto-phosphorylation of Aurora B/C on chromosome arms was unaltered in the mutant oocytes (***Figure 9B***). Importantly, Sgol2 did not alter the overall amount of Aurora C at kinetochores (***Figure 10***). These results imply that Sgol2 counters phosphorylation of KMN network proteins by a mechanism not involving its recruitment of PP2A. Strangely, this process seems to depend on Sgol2's prior phosphorylation by the very same kinases that it ultimately inhibits. In other words, Aurora B/C creates its own inhibitor at kinetochores, creating a negative feedback loop. Consistent with this notion, the gradual accumulation at kinetochores of Sgol2 phosphorylated at T521 following GVBD (***Figure 7B***) is accompanied by a corresponding reduction in Dsn1 Ser100 phosphorylation (***Figure 9D***). Critically, Sgol2's ability to inhibit Aurora B/C kinase activity at kinetochores does not depend on kinetochore-microtubule interactions, as Sgol2 reduced auto-phosphorylation of Aurora B/C even in the presence of microtubule destabilizing drug (***Figure 9C***). To address whether hyper-activity of Aurora B/C at kinetochores could be responsible for the paucity of K-fibers in $sgol2^{\Delta/\Delta}$ oocytes, we tested the effect of the Aurora kinase inhibitor AZD1152. This both increased the density of K-fibers and reduced kinetochore stretching but did not eliminate lateral attachments (***Figure 8B***), which may require direct involvement of MCAK. We suggest that Sgol2 promotes chromosome bi-orientation by inhibiting Aurora B/C kinase as well as by recruiting MCAK.

## Discussion

Previous work has shown that Sgol2 is essential for protection of centromeric cohesin from separase at the first meiotic division (***Lee et al., 2008***; ***Llano et al., 2008***). Our creation and analysis of mutations within Sgol2's highly conserved coiled coil reveals the likely mechanism, namely recruitment of PP2A to centromeres. PP2A might

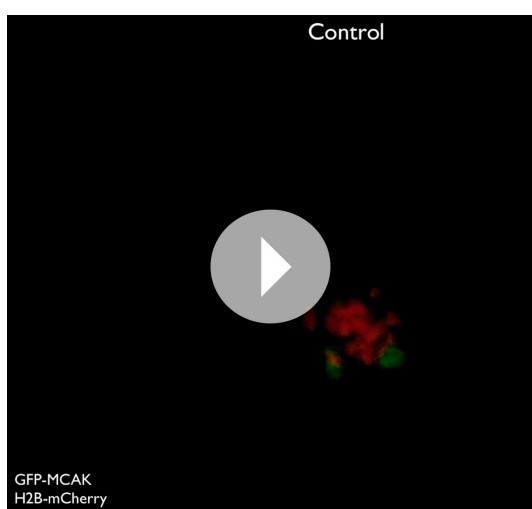

**Video 1**. Z-projected time-course confocal microscopy video of control oocyte microinjected at GV stage with GFP-MCAK and H2B-mCherry mRNA.

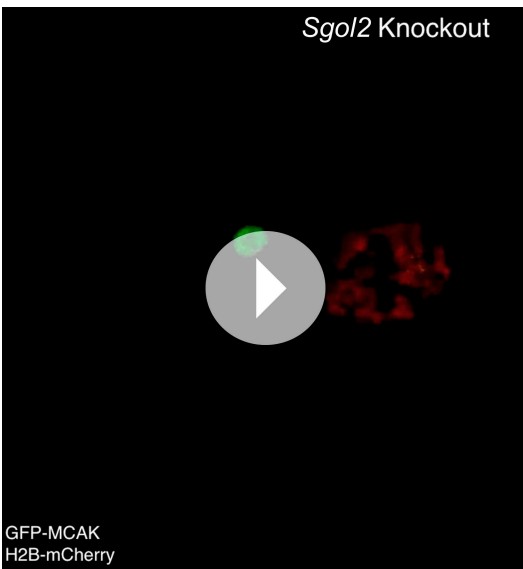

**Video 2**. Z-projected time-course confocal microscopy video of *sgol2^Δ/Δ* oocyte microinjected at GV stage with GFP-MCAK and H2B-mCherry mRNA.

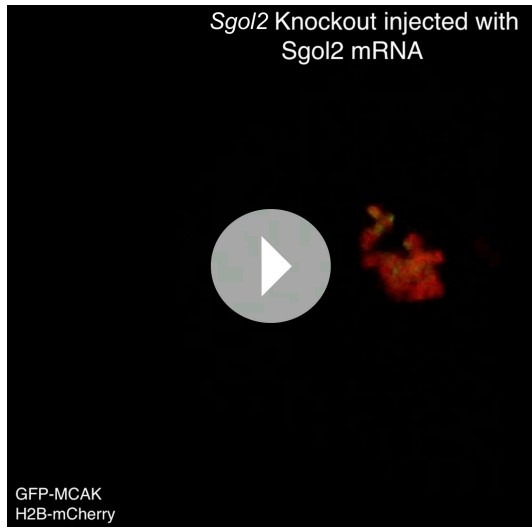

**Video 3**. Z-projected time-course confocal microscopy video of *sgol2^Δ/Δ* oocyte microinjected at GV stage with GFP-MCAK, H2B-mCherry and Sgol2 mRNA.

protect cohesin from separase either directly, by binding (*Holland et al., 2007*) and inhibiting separase activity, or indirectly, by reversing phosphorylation of cohesin's Rec8 subunit (*Katis et al., 2010*), a process known to be important for its cleavage in yeast. However, if cohesion protection were Sgol2's sole function, then its depletion should not compromise chromosome segregation during meiosis I. The precocious loss of cohesion should only affect the second meiotic division. On the contrary, depletion of Sgol2 specifically in oocytes has highly pleiotropic consequences already during meiosis I. These include inefficient SAC silencing, greatly increased bivalent stretching, and major alterations in the dynamics of kinetochore-microtubule attachments (K-fibers). We show that Sgol2 silences the SAC by binding to Mad2 using a similar MIM motif common to Mad2's other two partners Cdc20 and Mad1. This raises the possibility that Sgol2 silences the SAC by competing with Cdc20 for the binding of activated Mad2, thereby preventing formation of the MCC. However, this cannot be the full story as SAC silencing also depends on Sgol2's ability to bind PP2A. Because Sgol2's MIM motif is close to the coiled coil by which it binds PP2A, it is conceivable that PP2A directly (i.e., structurally) facilitates the formation or stability of 'closed' complexes between Sgol2 and Mad2. This hypothesis is supported by the fact that only dephosphorylated Mad2 interacts with Cdc20 and Sgol2, while a phosphomimetic mutant of Mad2 cannot associate with either of these proteins (personal communication Michael Orth). Thus, PP2A could selectively stabilize the Sgol2-Mad2 complex by keeping Mad2 in a de-phosphorylated state. Therefore, it is conceivable that if there were sufficient turnover of Mad2 bound to MCC, then PP2A could promote the Sgol2: Mad2 complex formation and thus alter the activity of the MCC responsible for inhibiting the APC/C.

Unlike centromeric protection and SAC silencing, Sgol2's regulation of bivalent stretching depends not on its ability to bind PP2A but instead on phosphorylation at T521 and T600 by Aurora B/C kinases. This facilitates MCAK accumulation at kinetochores. Because artificial recruitment of MCAK to kinetochores by fusion to CENP-B substitutes for Sgol2 with regard to bivalent stretching, we suggest that it is the population of MCAK recruited to kinetochores by Sgol2 that moderates the pulling forces exerted on maternal and paternal kinetochores, possibly by reducing the incidence of lateral attachments.

Artificial MCAK recruitment did not however restore normal looking end on attachments or greatly facilitate bi-orientation in Sgol2 deficient oocytes, implying that Sgol2 has yet another function at kinetochores besides recruiting PP2A and MCAK. Our observation that auto-phosphorylation of Aurora B/C kinases as well as phosphorylation of KMN proteins are elevated in Sgol2's absence but depressed upon microinjection of Sgol2 mRNA suggests that this other function of Sgol2 is to act

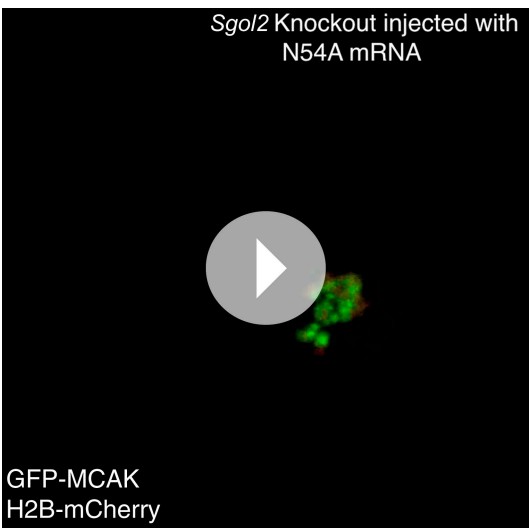

**Video 4**. Z-projected time-course confocal microscopy video of *sgol2^Δ/Δ* oocyte microinjected at GV stage with GFP-MCAK, H2B-mCherry and N54A Sgol2 mRNA.

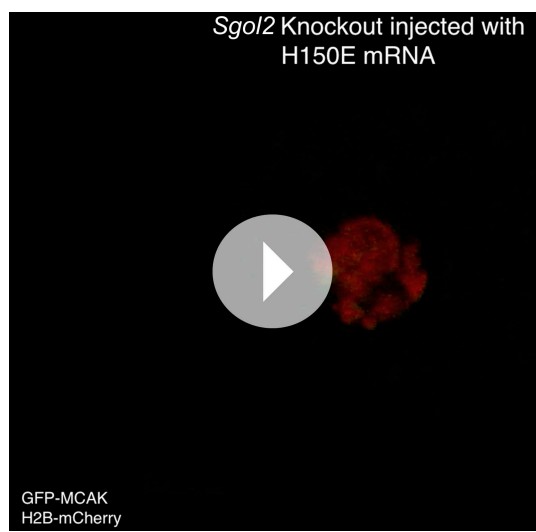

**Video 5**. Z-projected time-course confocal microscopy video of *sgol2^Δ/Δ* oocyte microinjected at GV stage with GFP-MCAK, H2B-mCherry and H150E Sgol2 mRNA.

as a kinetochore-specific inhibitor of Aurora B/C kinases. The notion that Sgol2 moderates Aurora B/C kinase activity at kinetochores was unexpected, as it has been reported that xSgo2, a newly discovered homolog of shugoshin in *Xenopus*, actually promotes Aurora Kinase activity, generating a favourable environment for microtubule polymerization (*Rivera et al., 2012*). In this case, the effect may be systemic, that is throughout the cytoplasm, whereas *M.m* Sgol2 appears to act as an inhibitor only at kinetochores. A remarkable feature of this effect is that Sgol2's ability to inhibit Aurora B/C kinases at kinetochores depends on its prior phosphorylation at T521 and T600 by the very same Aurora kinases. In other words, Aurora B/C kinase creates its own inhibitor, generating a negative feedback loop. Whether the phosphorylation of Sgol2 necessary for it to act as an inhibitor of Aurora kinases at kinetochores is also generated at this location or along chromosome arms or elsewhere in the cell is currently unclear.

Irrespective of its precise spatial and dynamic properties, this negative feedback loop may be an important feature of meiosis I. Aurora B/C protein kinases have a key role in destabilizing inappropriate kinetochore-microtubule attachments, namely those that do not generate tension (*Lampson and Cheeseman, 2011*). They are therefore essential for the efficient bi-orientation of bivalent chromosomes. However, it is important that the de-stabilization process is eventually turned off so that correctly attached bivalents remain stably attached to microtubules. Hitherto, it has been assumed that the mechanism responsible for turning off these kinases merely involved physical forces that pull Aurora substrates away from the enzymes when maternal and paternal kinetochores are pulled in opposite directions (*Liu et al., 2009*). Though our work does not preclude a role for traction in down regulating Aurora B/C's access to kinetochore-bound substrates, it raises the possibility that down regulation is also an intrinsic feature of a biochemical system in which Aurora B/C activates its own inhibitor. The regulatory network revealed by our work ensures that Aurora B/C kinases will be active early during the bi-orientation, when incorrect attachments may be more frequent, and less active later, when they have been largely eliminated. It is possible that during the period when Aurora B/C kinases are active, the selective stabilization of correct attachments is mediated by mechanical changes within kinetochores induced by tension, as recently observed in purified yeast kinetochores (*Akiyoshi et al., 2010*), not as originally thought by tension dependent changes in Aurora kinase activity, for which there has never been much direct evidence. We suggest that an important feature of meiosis I in oocytes is an early phase of high Aurora B/C kinase activity at kinetochores followed by a later phase of lowered activity made possible by the Aurora B/C-Sgol2 negative feedback loop. The lowered activity of Aurora B/C might increase the MT-depolymerizing activity of MCAK at kinetochores (*Andrews et al., 2004*). This is supported by the fact that the kinetochore pool of MCAK, recruited by Sgol2, is indeed not phosphorylated at serine 92 in mouse oocytes

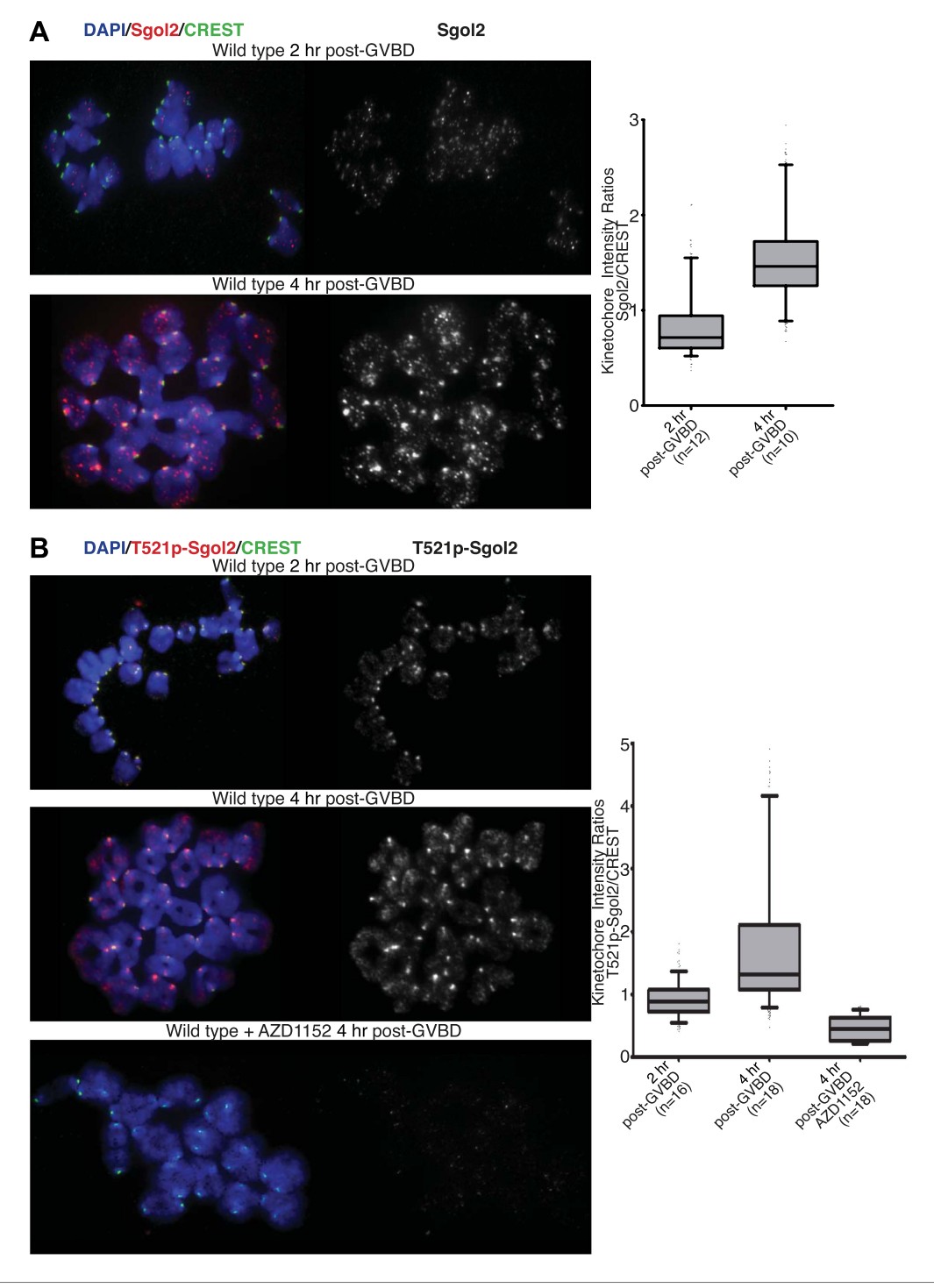

**Figure 7**. Aurora B/C Kinase phosphorylates Sgol2 on T521. (**A**) Oocytes harvested from wild type control females at GV stage were matured in vitro and chromosome spreads were prepared at 2 and 4 hr post-GVBD. Slides were stained for DNA (blue), Sgol2 (red), and CREST (green). Fluorescence intensity ratios of Sgol2 and CREST at kinetochores were quantified. Numbers of oocytes examined are indicated (n). (**B**) Chromosome spreads were prepared at 2 and 4 hr post-GVBD from wild type control oocytes. Slides were stained for DNA (blue), anti-T521p-Sgol2 (red), and CREST (green). Aurora B/C kinase activity was inhibited by culturing oocytes in M16 medium supplemented with the AZD1152 (100 nM) inhibitor. Fluorescence intensity ratios of T521p–Sgol2 and CREST at kinetochores were quantified.

(data not shown). If correct, it will be important to understand the dynamic properties of the feedback loop that ensures that Aurora B/C kinases do not turn themselves off precociously. It is currently unclear whether the Aurora B/C-shugoshin feedback loop is a general feature of meiotic cells or whether it is also a feature of mitotic cells. How Sgol2 inhibits Aurora B/C kinase activity at kinetochores is also unclear, in particular whether Sgol2 does this alone or only in conjunction with the recruitment of MCAK.

Given the number of fundamental processes regulated by Sgol2, it is surprising that *sgol2*$^{\Delta/\Delta}$ oocytes often manage to undergo the first meiotic division without generating massive aneuploidy. Their second meiotic division is of course completely defective due to the total absence of sister chromatid cohesion. We suggest that all of the processes regulated by Sgol2 are fundamentally essential but that they are also regulated by Sgol2-independent pathways. In other words, the processes regulated by Sgol2 may be so fundamental that there are multiple pathways regulating them. It is not inconceivable that Sgol1, which is also expressed during meiosis, shares some functions with Sgol2 and it will therefore be interesting to analyze the phenotype of double mutant oocytes.

What is possibly most remarkable is that a single protein regulates sister chromatid cohesion, chromosome traction, the spindle assembly checkpoint, and lastly Aurora B/C kinases. Crucially, by demonstrating that different point mutations alter the regulation of some but not other processes, we have proven that the pleiotropy observed in *sgol2*$^{\Delta/\Delta}$ oocytes is not due to knock on effects (secondary or even tertiary pathology) of eliminating a single function. Sgol2 possesses several different domains that regulate fundamentally different processes (*Figure 11*). Why so much regulation is embedded into a single protein is unclear, but we speculate that this helps the coordination of multiple processes necessary for high fidelity chromosome segregation.

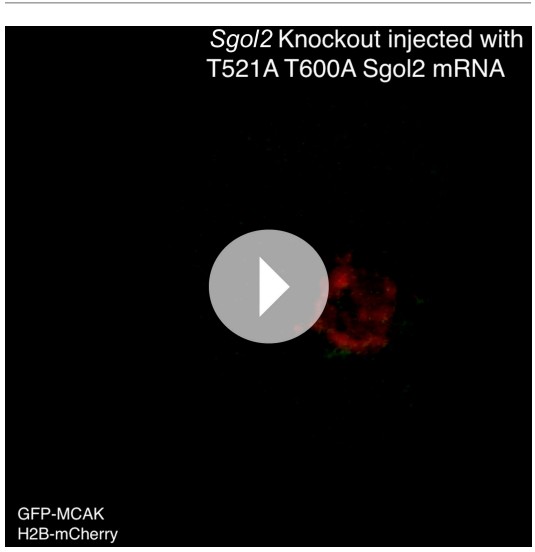

**Video 6**. Z-projected time-course confocal microscopy video of *sgol2*$^{\Delta/\Delta}$ oocyte microinjected at GV stage with GFP-MCAK, H2B-mCherry and T521A T600A Sgol2 mRNA.

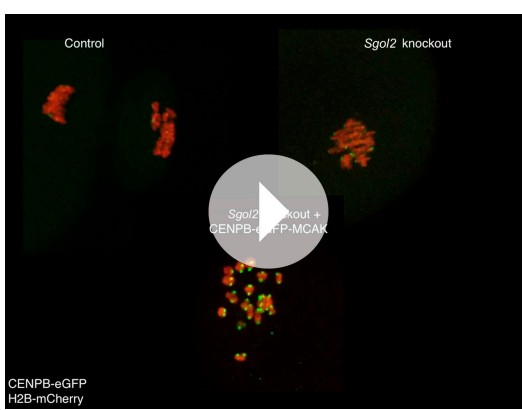

**Video 7**. Upper panel displays Z-projected time-course confocal microscopy video of control and *sgol2*$^{\Delta/\Delta}$ oocytes microinjected at GV stage with CENPB-eGFP and H2B-mCherry mRNA. Lower panel displays representative confocal video of *sgol2*$^{\Delta/\Delta}$ oocyte microinjected with CENPB-eGFP-MCAK and H2B-mCherry mRNA.

## Materials and methods

### Generation of mouse strains

Generation of the *Sgol2* knockout mice has been described (*Llano et al., 2008*). To detect Rec8, females from a transgenic line expressing Rec8 from a bacterial artificial chromosome (BAC), with nine tandem copies of the human c-myc epitope at its C terminus, were crossed to *sgol2*$^{\Delta/+}$ males. Litters with genotypes *sgol2*$^{\Delta/+}$ (Tg)*Rec8- Myc+* were crossed back to *sgol2*$^{\Delta/+}$ mice to generate *sgol2*$^{\Delta/+}$ (Tg)*Rec8-Myc+* (controls) and *sgol2*$^{\Delta/\Delta}$ (Tg)*Rec8-Myc+* female.

### Isolation and culture of oocytes

Prophase-arrested oocytes were harvested from ovaries excised from 8–12 week old female mice. Antral follicles were isolated using sterile insulin needles in M2 medium (Sigma-Aldrich, UK) supplemented with 200 µM IBMX (Sigma-Aldrich). For intracellular microinjections, oocytes were placed in droplets of M2 medium with IBMX. For time-lapse confocal imaging and in vitro culture prior

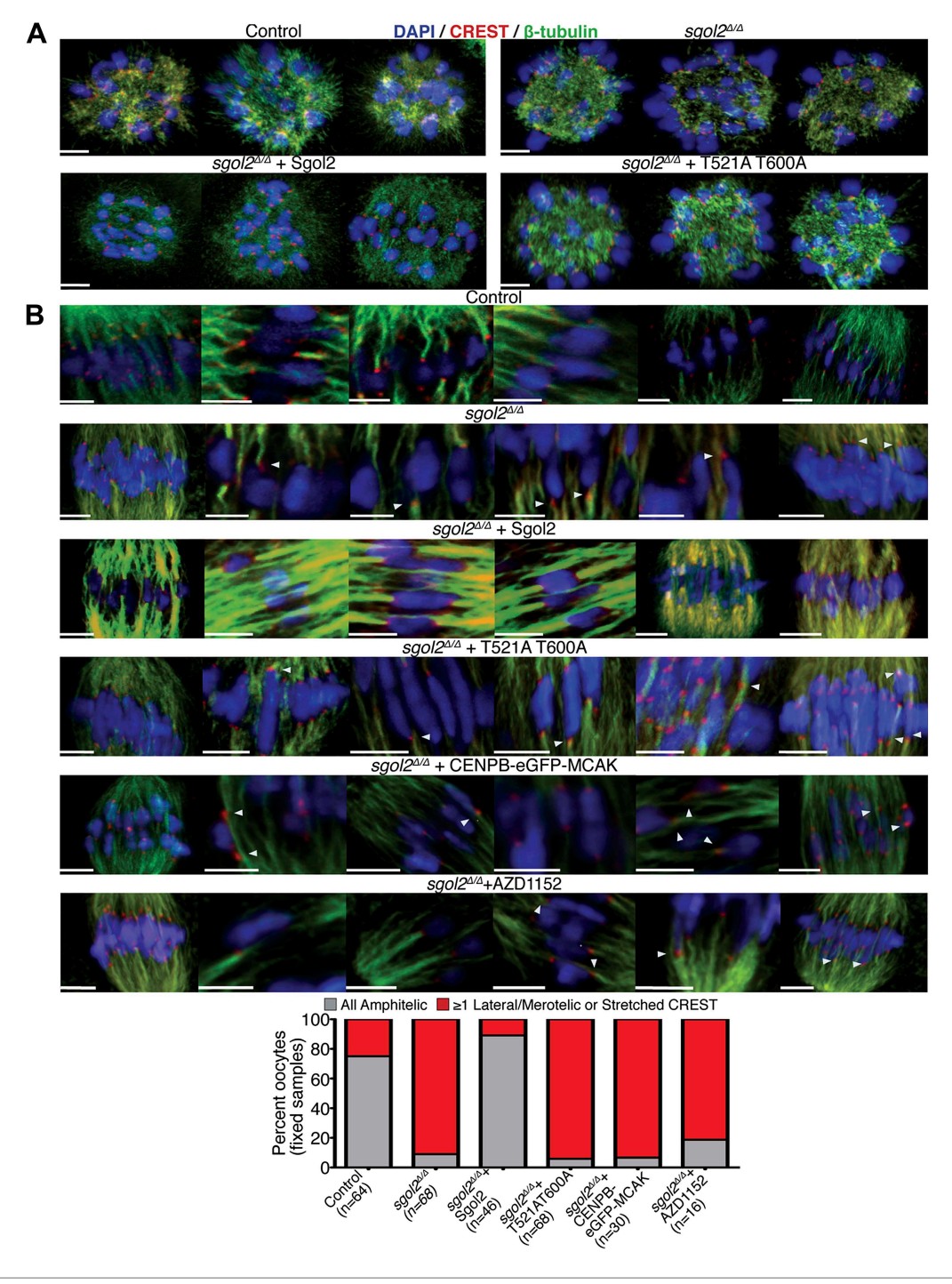

**Figure 8**. Sgol2 promotes bi-orientation of homologous chromosomes by increasing kinetochore-microtubule interactions. (**A**) To study the relative distributions of chromosomes on the ball shaped spindle, at 2–3 hr post-GVBD, oocytes were treated with Ca$^{2+}$ buffer for 90 s and then fixed in 1% PFA. Cells were stained with anti-β-tubulin (microtubules, green), CREST (kinetochores, red), and DAPI (chromosomes, blue). Scale bars represent 5 μm. (**B**) Oocytes harvested at GV stage from wild type, *sgol2*$^{Δ/Δ}$ and *sgol2*$^{Δ/Δ}$ microinjected with wild type, T521A T600A Sgol2 mRNA and CENPB-eGFP-MCAK mRNA were cultured in M16 medium for 6 hr. At 6 hr post-GVBD, oocytes were fixed in 1% PFA were stained with anti-β-tubulin (microtubules, green), CREST (kinetochores, red), and DAPI (chromosomes, blue). For Aurora B/C inhibition, oocytes harvested from *Sgol2* knockout females were cultured in M16 medium for 4 hr. At 4 hr post-GVBD, oocytes were transferred to M16 medium supplemented with low concentration (20 nM) of AZD1152 inhibitor. Oocytes were cultured until 6 hr post-GVBD and were then fixed in 1% PFA. Scale bars represent 5 μm.

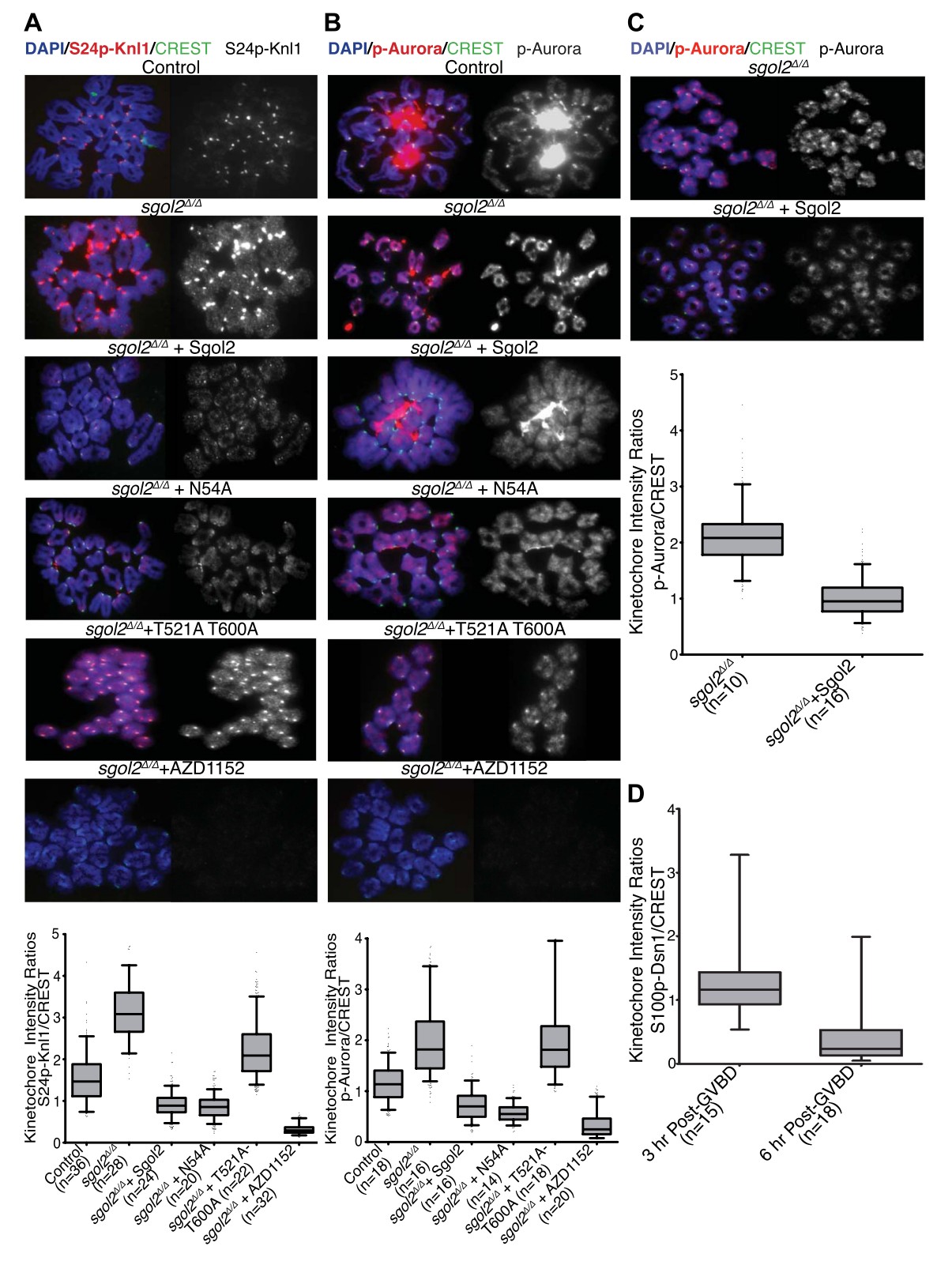

**Figure 9**. Sgol2 inhibits Aurora B/C Kinase activity at kinetochores. (**A**) Chromosome spreads were prepared from indicated groups at 4 hr post-GVBD. Slides were stained with DAPI (blue), anti-S24p-Knl1 (red) and CREST (green). Kinetochore intensities of anti-S24p-Knl1 and CREST signals were quantified and the intensity ratios of anti-S24p-Knl1 and CREST signals are shown. Upper and lower bars indicate 95th and 5th percentiles, respectively. (**B**) Chromosome spreads

*Figure 9. Continued on next page*

Figure 9. Continued

prepared at 4 hr post-GVBD were stained with DAPI (blue), anti-p-Aurora A/B/C (red) and CREST (green). Kinetochore fluorescence intensity ratios of p-Aurora A/B/C and CREST are displayed. (**C**) *Sgol2* knockout and *Sgol2* knockout oocytes injected with wild type Sgol2 mRNA were cultured for 6 hr from GVBD in M16 medium supplemented with 400 nM Nocodazole. Chromosome spreads were stained for DNA (DAPI, blue), p-Aurora A/B/C (red), and CREST (green). (**D**) anti-S100p-Dsn1 and CREST kinetochore intensity ratios, quantified from chromosome spreads prepared at 3 and 6 hr post-GVBD from wild type control oocytes.

to preparation of chromosome spreads, oocytes were transferred into IBMX-free M16 medium and cultured at 37°C and 5% $CO_2$.

## Preparation of mRNAs

Capped mRNA constructs with a poly-A tail were transcribed using T3 or T7 Ultra mMESSAGE kit (Ambion, Austin, TX, USA) from plasmid cDNA encoding H2B-mCherry, wild type Sgol2 (Origene, Rockville, MD, USA), and various mutants of Sgol2, GFP-Sgol2, Securin-eGFP, GFP-MCAK, CENPB-eGFP, CENPB-eGFP-MCAK, and GFP-EB3.

## Microinjection of mRNA

Fully-grown oocytes were injected with 5–10 pl mRNA at a final concentration of 0.1 mg/ml in RNase-free water (Ambion) using a Pneumatic PicoPump (World Precision Instruments). To allow for protein expression, following microinjection of mRNA, GV oocytes were cultured for 12 hr in M16 supplemented with IBMX. Oocytes were then washed in inhibitor-free M16 and thereafter cultured at 37°C and 5% $CO_2$.

## Live cell confocal microscopy

During time-lapse confocal microscopy experiments, oocytes were cultured in a PeCon environmental microscope incubator at 37°C and 5% CO2. A Zeiss LSM510 META confocal microscope equipped with PC-Apochromat 63x/1.2 NA water immersion and PC-Apochromat 20x/0.8 NA objective lenses was used for image acquisition. For detection of GFP and mCherry, 488-nm and 561-nm excitation wavelengths and BP 505–550 and LP 575 filters were used. During live-cell imaging, chromosomes labeled with H2B-mCherry were tracked with an EMBL-developed tracking macro (*Rabut and Ellenberg, 2004*) adapted to our microscope. We imaged 8 to 16 z-confocal sections (every 1.5 or 2.0 μm) at 5–10 min intervals for 12–14 hr.

## Chromosome spreads

Chromosome spreads from mouse oocytes were prepared as previously described (*Peters et al., 1997*; *Hodges and Hunt, 2002*). Briefly, zona pellucida was removed by treatment with 10 mg/ml Pronase (Sigma-Aldrich) at 37°C for 5–10 min. Zona-free oocytes were transferred to hypotonic solution (50% FCS in deionized water) for 10 min at 37°C. Oocytes were then fixed in paraformaldehyde solution (1% paraformaldehyde, 0.15% Triton X-100, 3 mM dithiothreitol, adjusted to pH 9.2 with NaOH) in 15-well multichamber glass slide (MP Biomedicals) overnight at room temperature in a humidified chamber. After drying, slides were washed twice in 0.4% Photoflo/$H_2O$ (Kodak) and PBS for 5 min, at room temperature. Slides were then processed for immunostaining. In this study, we used CREST (1:250; Davis Lab, Davis, CA, USA), rabbit anti-Sgol2 serum (1:50; gift from JL Barbero [*Parra et al., 2009*]), anti-phospho T521 Sgol2 (1:500, gift from Yoshi Watanabe [*Tanno et al., 2010*]), anti-MCAK (1:500, gift from Duane A Compton [*Mack and Compton, 2001*]), anti-phospho S24 Knl1 (1:1000, gift from Iain Cheeseman [*Welburn et al., 2010*]), anti-phospho S100 Dsn1 (1:500, gift from Iain Cheeseman [*Welburn et al., 2010*]), antibody detecting T288, T232 or T198 on Aurora A/B/C (1:50, Cell Signaling Technology, Danvers, MA, USA), anti-Aurora C (1:50, Bethly Laboratories, Montgomery, TX, USA), mouse anti-PP2A-C (1:200, BD Biosciences, Palo Alto, CA, USA), and anti-c-Myc (1:500, Millipore, Billerica, MA, USA).

## Immunofluorescence

For analysis of K-fibers, oocytes were processed as previously described (*Kitajima et al., 2011*). Briefly, after either 2 or 6 hr post-GVBD, oocytes were treated for 90 s with $Ca^{2+}$ buffer (100 mM PIPES pH 7:0, 1 mM $MgCl_2$, 0.1 mM $CaCl_2$, 0.1% Triton X-100) at 37°C and then fixed in 1% formaldehyde prepared in $Ca^{2+}$ buffer for 30 min at room temperature. Following fixation, oocytes were washed and extracted overnight in PBT (PBS supplemented with 0.1% Triton X-100) at 4°C. Cells were then blocked in 3% BSA with PBT (blocking solution) for 1 hr at room temperature. For immuno-labeling oocytes were incubated overnight at 4°C with β-tubulin (1:250, Abcam, Cambridge, UK) and CREST (1:100, Davis Lab, Davis, CA, USA)

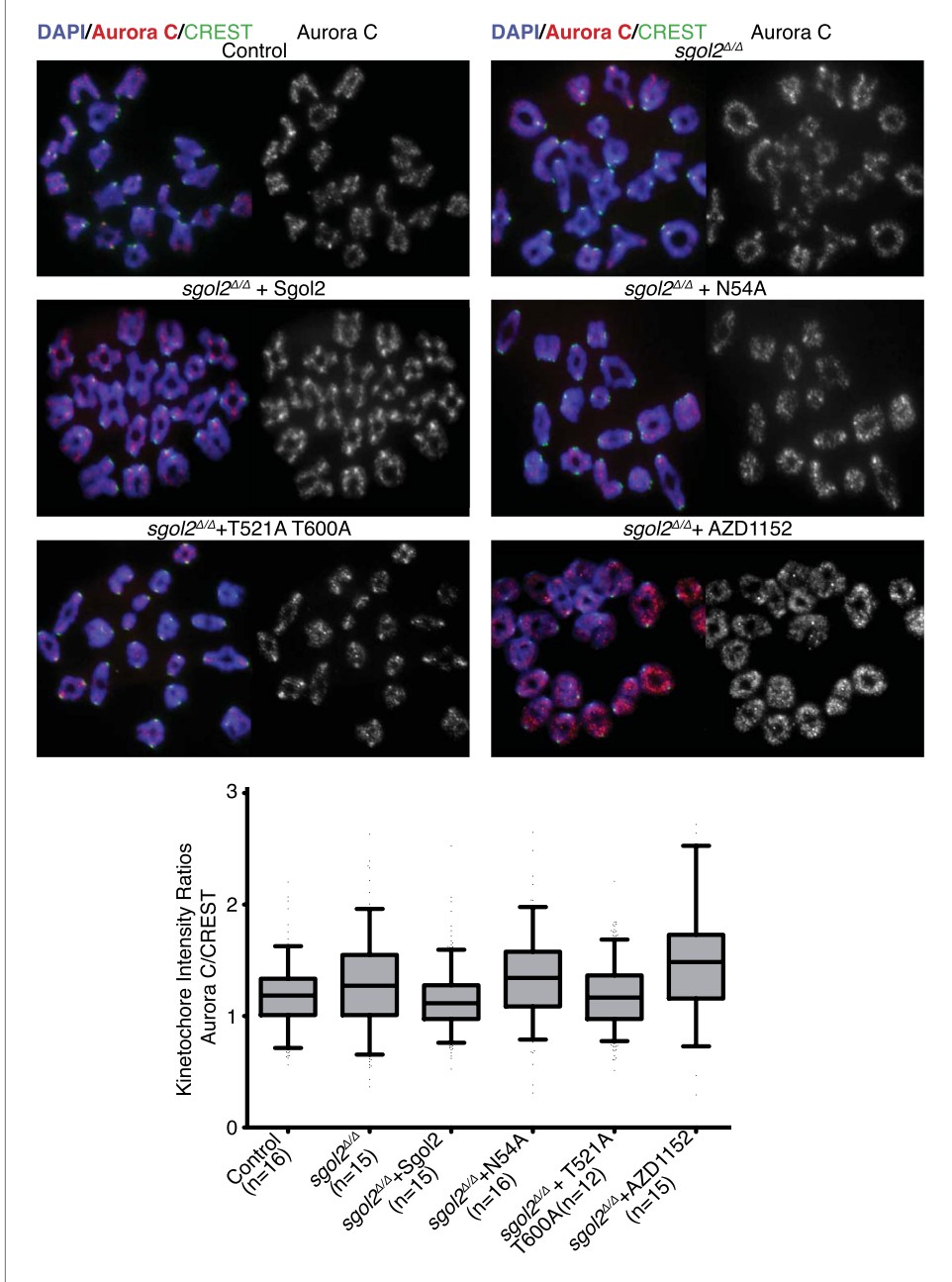

**Figure 10**. Sgol2 decreases Aurora C activity at the kinetochores without affecting its localization. Chromosome spreads were prepared from the indicated groups at 4 hr post-GVBD. Slides were stained for DNA (DAPI, blue), Aurora C (red), and CREST (green). Fluorescence intensity ratios of Aurora C and CREST at kinetochores are displayed. Upper and lower bars indicate 95th and 5th percentiles, respectively.

antibodies, prepared in blocking solution. After four washes in PBT, oocytes were incubated for 1 hr at room temperature with Alexa 488 and 568 conjugated secondary antibodies (1:500, Invitrogen, UK) prepared in blocking solution. Oocytes were briefly stained with Hoechst 33,342 (20 mg/ml) before confocal imaging.

## Protein expression and purification
Recombinant MBP-Sgol2cc-His and MBP-Sgol2cc-Flag were purified from 1l of *Escherichia coli* BL21(DE3) RIPL (Stratagene, La Jolla, CA, USA) carrying corresponding expression cassettes in

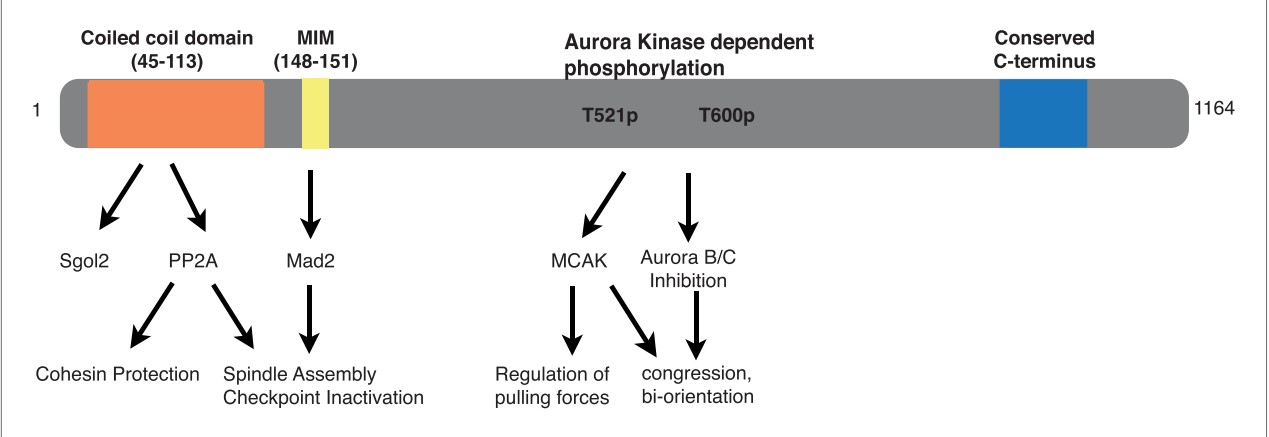

**Figure 11**. Graphical summary of sequence features, interactions and key functions of Sgol2 in mouse oocytes.

pMAL-c2x- (Flag-tagged subunit) and pET28b-based backbones (His-tagged subunit), grown at 37°C in 2xTY media supplemented with 25 µg/ml chloramphenicol and 100 µg/ml ampicillin for pMAL-c2x or 50 µg/ml kanamycin for pET28b. At $OD_{600}$ = 0.4, MBP-Sgol2cc-His or -Flag synthesis was induced at 30°C by the addition of 0.1 mM IPTG. After 4 hr, the cells were harvested by centrifugation, frozen in dry ice, and stored at −80°C. The cell paste was re-suspended in 40 ml of lysis buffer (50 mM sodium phosphate pH 7.0, 500 mM NaCl, 10% glycerol, 0.02% Triton X-100) containing Roche complete EDTA free protease inhibitor, 1 mM PMSF and, for the His-tagged subunit 5 mM. Each aliquot was sonicated four times for 30 s on ice. Insoluble material was removed by centrifugation at 25,000 rpm for 1 hr at 4°C in a Beckman Coulter Avanti-J26XP JLA8.1000 rotor. The supernatant was incubated for 2 hr at 4°C on a 5 ml Talon column (BD Biosciences, Palo Alto, CA, USA) or ANTI-FLAG M2 Affinity Gel (Sigma A2220). The M2 Flag beads were washed four times with 50 ml of lysis buffer and the Talon beads were washed two times with 50 ml of lysis buffer containing 20 mM imidazole and two times with 50 ml of lysis buffer containing 45 mM imidazole. MBP-Sgol2cc-Flag was eluted with four times 1 ml of lysis buffer containing 3 × FLAG peptide (150 ng/µl). MBP-Sgol2cc-His was eluted with 10 ml of lysis buffer containing 1 M of imidazole. The concentration of MBP-Sgol2cc-His and MBP-Sgol2cc-Flag were determined by nanodrop against the lysis solution.

## PP2A-Sgol2 In vitro binding assay

For all binding assays, highly-purified PP2A subunits (purified as previously described [*Cho and Xu, 2007*]) and MBP-Sgol2cc-Flag were used. To test the binding of PP2A to mutant shugoshin, 20 µg of GST-tagged PP2A A subunit were incubated with excess amounts of PP2A B56- and C- subunit and MBP-tagged shugoshin for 1 hr on ice. Subsequently, samples were incubated with 40 µl of glutathione beads (GE Healthcare) for 20 min. Beads were then washed 4 × with 1 ml each of 1 × PBS, 0.1% Triton X-100, 1 mM dithiothreitol, boiled in reducing SDS loading buffer, and subjected to SDS-PAGE analysis followed by Coomassie blue staining.

## Co-immunoprecipitation of Sgol2 coiled coil domains

His and Flag-tagged mSgol2cc from wild type and 3A were co-expressed in *E.coli* BL21(DE3)RIPL for 4 hr at 30°C with 0.1 mM of IPTG. Cells were spun at 25,000 rpm for 1 hr at 4°C in a Beckman Coulter Avanti-J26XP JLA8.1000 rotor. The pellet was re-suspended into lysis buffer (50 mM sodium phosphate pH = 7; 250 mM NaCl; 10% glycerol, 0.02% Triton X-100; protease inhibitor). 10 mg of extract were incubated with 100 µl of Talon beads for 2 hr at 4°C. The beads were wash five times with the lysis buffer containing 23 mM of imidazole. MBP-Sgol2cc-His was eluted with 10 ml of lysis buffer containing 1 M of imidazole. Affinity purified material and input were resolved by SDS-PAGE. The abundance of His and Flag-tagged MBP-Sgol2cc were analyzed by western blot using anti-Flag (1:10,000) and anti-His (1:10,000) antibodies.

## Acknowledgements

We would like to thank members of the Nasmyth lab, Bela Novak, and Elwy Okaz for discussions and advise, and Melina Schuh for critical reading of the manuscript. We thank Lysie Champion and

Kikue Tachibana-Konwalski for the pRNA-CENPB-eGFP vector, and Jan Ellenberg for sharing the pGEMHE-EB3-eGFP construct. We are grateful to Iain Cheeseman, Arshad Desai, Duane A Compton, JL Barbero, Jason Swedlow, Yoshi Watanabe, and Linda Wordeman for providing antibodies. We would like to thank Elena Llano for providing the *Sgol2* targeted embryos. We would also like to thank Chris Preece and Ben Davies for valuable microinjection training, Michael Orth and Bulent Cetin for sharing their unpublished findings, and animal care staff at the BSB Facility, Dept. of Biomedical Services for technical assistance. AR was supported by a PhD fellowship from the Boehringer Ingelheim Fonds. BM and OS were supported by a grant of the Deutsche Forschungsgemeinschaft (STE997/3-2 within the priority program SPP1384). AP was supported by SAF2011-25252. This study was funded by the European Community's Seventh Framework MitoSys/241548, Medical Research Council and Wellcome Trust.

## Additional information

### Funding

| Funder | Grant reference number | Author |
|---|---|---|
| Wellcome Trust | 019859/Z/10/Z | Kim Nasmyth |
| Medical Research Council | 84673 | Kim Nasmyth |
| Boehringer Ingelheim Fonds | | Ahmed Rattani |
| Deutsche Forschungsgemeinschaft | STE997/3-2 within the priority program SPP1384 | Olaf Stemmann |
| Ministerio de Ciencia e Innovacion | SAF2011-25252 | Alberto Pendas |
| European Community's Seventh Framework Programme | MitoSys/241548 | Kim Nasmyth |

The funders had no role in study design, data collection and interpretation, or the decision to submit the work for publication.

### Author contributions

AR, Conception and design, Acquisition of data, Analysis and interpretation of data, Drafting or revising the article; MW, MP, WH, SM, JG, Acquisition of data, Analysis and interpretation of data; BM, Acquisition of data, Analysis and interpretation of data, Drafting or revising the article; WX, Analysis and interpretation of data, Contributed unpublished essential data or reagents; OS, Drafting or revising the article, Contributed unpublished essential data or reagents; AP, KN, Conception and design, Drafting or revising the article

### Ethics

Animal experimentation: Animal studies were carried out under the UK Home Office ASPA project licence 30/2824.

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
