## [Decision Letter]

Thank you for sending your work entitled “Sgol2 provides a regulatory platform that coordinates essential cell cycle processes during meiosis I in oocytes” for consideration at *eLife*. Your article has been favorably evaluated by a Senior editor and 3 reviewers, one of whom is a member of our Board of Reviewing Editors.

The Reviewing editor and the other reviewers discussed their comments before we reached this decision, and the Reviewing editor has assembled the following comments to help you prepare a revised submission.

We all were favourably impressed by your systematic analysis of the surprisingly numerous functions of Sgol2 in chromosome segregation. You demonstrate that PP2A localization and protection of centromeric cohesion depends on Sgol2, and more specifically, on the coiled-coil domain of Sgol2, and provide here the final proof that PP2A localization through Sgol2 brings about centromeric cohesin protection in mammalian oocyte meiosis I.

1) There was some discussion among the reviewers as to whether it would be important to show the localization of a component of the SAC – is Mad2 remaining at kinetochores in Sgol2 knockout oocyte? On balance we decided that it was not essential, but if you have the data it would strengthen the message.

2) All the reviewers felt you could improve the citation of previous work in the Introduction. In particular, you should provide a citation for the statement that stretching the bivalents causes terminalisation of chiasmata. Furthermore, you should discuss the claim that stretching bivalents causes terminalisation without any apparent loss of cohesin.

---

## [Author Response]

*1) There was some discussion among the reviewers as to whether it would be important to show the localization of a component of the SAC – is Mad2 remaining at kinetochores in Sgol2 knockout oocyte? On balance we decided that it was not essential, but if you have the data it would strengthen the message*.

We tried to check if Mad2 residence is affected in Sgol2 knockout oocytes. We performed chromosome spreads at multiple time points, but unfortunately none of the three different antibodies we tested gave us a reliable signal that we could quantify. In short, at this time we do not have data for Mad2 levels in Sgol2 knockout oocytes.

*2) All the reviewers felt you could improve the citation of previous work in the Introduction. In particular, you should provide a citation for the statement that stretching the bivalents causes terminalisation of chiasmata. Furthermore, you should discuss the claim that stretching bivalents causes terminalisation without any apparent loss of cohesin*.

We do agree that this phrase could create confusion. Therefore in the revised version, we have removed the statement that stretching of the bivalents causes terminalisation of chiasmata from the Introduction and in the Results we have also modified the phrasing. We have described in the Results that though the bivalent in Sgol2 knockout oocytes appeared terminalised, they still retained cohesion between homologous chromosomes. Furthermore, we have added that this abnormal shape of bivalent chromosomes on chromosome spreads could possibly be explained by increased inter-kinetochore stretching on the proximal end of the chromosomes in Sgol2 knockout oocytes, but these bivalent chromosomes are still connected at distal ends by cohesin molecules.